

# Improved Cloud Phase Determination of Low Level Liquid and Mixed Phase Clouds by Enhanced Polarimetric Lidar

Robert A. Stillwell[1,2], Ryan R. Neely III[3,4], Jeffrey P. Thayer[2], Matthew D. Shupe[5,6], and David D. Turner[6]

[1]Advanced Study Program, National Center for Atmospheric Research, 3450 Mitchell Lane, Bldg 1, CO 80301, USA.
[2]Aerospace Engineering Sciences, University of Colorado at Boulder, ECNT 320, 431 UCB, University of Colorado, Boulder, CO 80309.
[3]School of Earth and Environment, University of Leeds, LS2 9JT, Leeds, UK.
[4]National Centre for Atmospheric Science, University of Leeds, LS2 9JT, Leeds, UK.
[5]Cooperative Institute for Research in Environmental Sciences, University of Colorado at Boulder, 216 UCB, University of Colorado, Boulder, CO 80309.
[6]Earth System Research Laboratory, National Oceanic and Atmospheric Administration, 325 Broadway, Boulder, CO 80309.

*Correspondence to:* Robert Stillwell (robert.stillwell@colorado.edu)

**Abstract.** The unambiguous retrieval of cloud phase from polarimetric lidar observations is dependent on the assumption that only cloud scattering processes affect polarization measurements. A systematic bias of the traditional lidar depolarization ratio can occur due to a lidar system's inability to accurately measure the entire backscattered signal dynamic range, and these biases are not always identifiable in traditional polarimetric lidar systems. This results in a misidentification of liquid water in clouds

5    as ice, which has broad implications on evaluating surface energy budgets. The Clouds Aerosol Polarization and Backscatter Lidar at Summit, Greenland employs multiple planes of linear polarization, and photon counting and analog detection schemes, to self evaluate, correct, and optimize signal combinations to improve cloud classification. Using novel measurements of diattenuation that are sensitive to both horizontally oriented ice crystals and counting system non-linear effects, unambiguous measurements are possible by over constraining polarization measurements. This overdetermined capability for cloud phase

10    determination allows for system errors to be identified and quantified in terms of their impact on cloud properties. It is shown that lidar system dynamic range effects can cause errors in cloud phase fractional occurrence estimates on the order of 30% causing errors in attribution of cloud radiative effects on the order of 10%-30%. This paper presents a method to identify and remove lidar system effects from atmospheric polarization measurements and uses co-located sensors at Summit to validate this method.


# 1 Introduction

Changing Arctic conditions lead to many changes in regional surface energy and mass budgets, which have a profound impact on humans outside the region (Curry et al., 1996; Hansen et al., 2011). Locked within the Greenland Ice Sheet (GrIS) is the the potential for sea level rise on the order of 7 m (Gregory et al., 2004), of which approximately 25 mm has already been

contributed from 1900 to present with an increased rate of mass loss in recent years (Kjeldsen et al., 2015). Several studies have linked variability of the surface energy and mass budgets to cloud properties and in particular low-level, liquid-only and mixed-phase[1] clouds (Bennartz et al., 2013; Sherwood et al., 2014; Miller et al., 2015; Tan et al., 2016; Miller et al., 2017). The climate is sensitive to Arctic cloud macro and microphysical properties, yet substantial gaps are present in understanding of fundamental cloud processes due to a limited set of cloud observations to which model results may be compared (Curry

et al., 1996; Cesana et al., 2012; Morrison et al., 2012; Bennartz et al., 2013; Van Tricht et al., 2016).

Understanding the nature of liquid-only and mixed-phase clouds is important for understanding the surface energy budget. Mixed-phase clouds show remarkable persistence in the Arctic even though the liquid phase is colloidally unstable, possibly persisting for days to weeks given the correct synoptic conditions (Shupe et al., 2006). Furthermore, though liquid-only and mixed-phase clouds can be found up to heights of approximately 6 km above mean sea level (amsl) in the Arctic, they have

been found by many to be predominately low-lying with high optical thickness[2] (Curry et al., 1996; Intrieri et al., 2002; Turner, 2005; Shupe et al., 2006; de Boer et al., 2009; Shupe, 2011; Shupe et al., 2013). Such characteristics make these clouds particularly hard to measure accurately from both the ground and space. Shupe et al. (2006) further notes that mixed-phase clouds are an understudied component of global cloudiness resulting in their poor representation in models at all scales, a finding supported by others including Cesana et al. (2012); Pithan et al. (2014); Kay et al. (2016). The focus of this work is the

interpretation of ground based polarimetric lidar measurements of Arctic liquid-only and mixed-phase clouds and assessing systematic measurement biases that inhibit their proper identification. While the scope of this work is confined to the Arctic, this work is informative to measurements of similar cloud types, for example present in Antarctic.

Polarimetric lidar systems are widely deployed to the polar regions to measure cloud properties. Nott and Duck (2011) and references therein summarize more than a dozen lidar deployment sites in the Arctic and Antarctic. Polarimetric lidar data

is particularly useful for cloud and aerosol studies to determine properties such as cloud phase, cloud base height, particle orientation, and for broad aerosol classifications (Schotland et al., 1971; Measures, 1984; Sassen, 1991; Kaul et al., 2004; Fujii and Fukuchi, 2005; Weitkamp, 2005; Freudenthaler et al., 2009; Hayman and Thayer, 2012; Groß et al., 2015). The utility of lidar observations can be enhanced by using complementary measurements that grant a more complete perspective such as cloud radars, microwave radiometers, and radiosondes as done for programs like the Surface Heat Budget of the Arctic

Ocean (SHEBA) (Shupe et al., 2006), the Department of Energy Atmospheric Radiation Measurement (ARM) program's

---

[1]This work uses the definition of mixed-phase presented by Shupe et al. (2008) where a mixed phase cloud is defined as a cloud system containing both liquid and ice water that interact via microphysical processes. The complete system must contain both liquid and ice water but no requirement is made on the exact location or quantity of either phase.

[2]In this manuscript, high is taken relative to ice-only clouds existing in the same region and not to liquid clouds existing in the mid-latitude or tropical regions



atmospheric observatories (Verlinde et al., 2016), and Mixed Phase Arctic Clouds Experiment (MPACE) (Verlinde et al., 2007). Despite its utility, polarimetric lidar has limitations. Among them is the stringent requirement of linear signal operation over a large dynamic range. If not properly designed or considered, measurements can be misinterpreted casting doubt on critical measurements like cloud phase (Hayman and Thayer, 2009; Liu et al., 2009; Neely et al., 2013). For example, traditional two-

channel orthogonal polarization measurements using co-polarized and cross-polarized signals can not unambiguously separate systematic polarization effects and geophysical effects. These measurement errors result in cloud phase misidentification, which, in turn, introduce unquantified errors into observationally based understanding of key cloud and radiative processes. Observations by lidar of Arctic liquid-only and mixed-phase clouds in particular are challenging due to their high optical thicknesses, relative to ice-only clouds, and low-lying altitude, which demands large system dynamic ranges.

This work focuses on novel polarimetric lidar measurements made at Summit, Greenland ($72°35'46.4"N$, $38°25'19.1"W$, $3212\,m\,amsl$) as part of the Integrated Characterization of Energy, Clouds, Atmospheric State, and Precipitation at Summit (ICECAPS) program outlined by Shupe et al. (2013). The primary measurements to be presented are taken from the Clouds Aerosol Polarization and Backscatter Lidar (CAPABL), which was originally designed to measure polarization properties of clouds with emphasis on identifying horizontally oriented ice crystals (HOIC) and cloud phase (Neely et al., 2013). Analysis

of seven years of polarimetric lidar data observed by CAPABL has highlighted several uncertainties and biases that can cause errors in the interpretation of geophysical retrievals of cloud phase, primarily caused by systemic limitations to adequately observe the dynamic range in backscattered signals from clouds.

The outline of this paper is as follows. The measurement theory, upon which the retrievals within CAPABL's automatic processing are based, is given in Sect. 2. An overview of the data collection and processing is provided in Sect. 3 with emphasis

on geophysical retrievals and potential errors caused by limited signal dynamic range. Several retrieval methods are presented and combined into a best estimate cloud identification in Sect. 4. A validation of the best estimate data product is presented in Sect. 5 using co-located micro-pulse lidar (MPL), microwave radiometer (MWR), millimeter cloud radar (MMCR), and broadband radiation measurement suite. Finally, this paper concludes with a discussion in Sect. 6 describing applicability of the presented observational methodology to other polar lidar measurements and quantification of lidar classification errors on

radiation budget estimates.

## 2 Measurement Theory

### 2.1 Polarization Measurements and Mueller Formalism

Polarimetric lidar leverages the vector nature of light to more completely describe scattering. Using a vector description of light allows one to describe scatterers by how they alter polarization states of light as well as how much energy is redirected.

Hayman and Thayer (2012) use polar decomposition of Mueller matrices to define the Stokes vector lidar equation (SVLE),





which links transmitted and received polarization states of light to physical attributes of the scatterers. This equation forms the basis of CAPABL's polarization retrievals and is given in Eq. 1

$$\bar{N}(R) = \bar{\bar{O}} \bar{\bar{M}}_{R_x}\left(\bar{k}_s\right) \left[ \left( G(R) \frac{A}{R^2} \Delta R \right) \bar{\bar{T}}_{atm}\left(\bar{k}_s, R\right) \bar{\bar{F}}\left(\bar{k}_i, \bar{k}_s, R\right) \bar{\bar{T}}_{atm}\left(\bar{k}_i, R\right) \bar{\bar{M}}_{T_x}\left(\bar{k}_i\right) \bar{S}_{T_x} + \bar{S}_B(\lambda_{R_x}) \right] \tag{1}$$

where $\bar{N}$ is vector of photon counts for each polarization channel as a function of range, $R$, $\bar{\bar{O}}$ is the observation matrix describing each polarization observation channel, $\bar{\bar{M}}_{T_x}$ and $\bar{\bar{M}}_{R_x}$ are the Mueller matrices describing the transmitter and receiver, which are functions of the incident and scattered wave vector $\bar{k}_i$ and $\bar{k}_s$, respectively, $G$ is the physical overlap function of the transmitter and receiver, $A$ is the telescope area, $\Delta R$ is the range resolution of the counting system, $\bar{\bar{T}}_{atm}$ is the one way transmission Mueller matrix either between the transmitter and the scatterer or between the scatterer and the receiver, $\bar{\bar{F}}$ is the scattering phase matrix, which is a function of both transmitted and received wave vectors and range, $\bar{S}_{T_x}$ is the Stokes vector of the light from the laser source, and $\bar{S}_B$ is the Stokes vector of the background condition which is a function of the receiver wavelength window, $\lambda_{R_x}$. The terms of the equation are organized by their functional order because matrix operations do not generally commute. The observation matrix is also included because only intensity can be measured directly with the full Stokes vector determined through measurement with particular configurations of the analyzer (Hayman and Thayer, 2012). For more information on the SVLE and its derivation, the reader is referred to Hayman and Thayer (2012).

Elements of $\bar{\bar{F}}$ can be used to describe physical attributes of scatterers beyond simple scattering cross section (Van De Hulst, 1957; Mishchenko and Hovenier, 1995; Kaul et al., 2004). The reader is referred to Neely et al. (2013) who describe the polarization retrievals and the physical interpretation of the elements CAPABL measures in detail. Here the retrieval presented by Neely et al. (2013) is generalized by relaxing the assumption made in that work that the receiver orientations are fixed at $0°$, $45°$, and $90°$ relative to the output linear polarization.

From this general form given in Eq. 1, the number of photons to be observed in any arbitrary linear polarization channel can be derived. Assuming that CAPABL: 1) emits a linear polarized signal at angle $\phi$, yielding the simplification

$$\bar{\bar{M}}_{T_x}\left(\bar{k}_i\right) \bar{S}_{T_x} = \begin{bmatrix} 1 & \cos\left(2\phi\right) & \sin\left(2\phi\right) & 0 \end{bmatrix}^T,$$

2) only measures linear polarized signal at angle $\theta$ from the reference transmit polarization, (Neely et al. (2013) Eq. 15 with $A\left(\Gamma_{wp}\right) = \bar{\bar{M}}_{R_x}(2\theta)$) yielding the simplification

$$\bar{\bar{M}}_{R_x}\left(\bar{k}_s\right) = \frac{1}{2} \begin{bmatrix} 1 & \cos\left(2\theta\right) & \sin\left(2\theta\right) & 0 \\ 1 & \cos\left(2\theta\right) & \sin\left(2\theta\right) & 0 \\ 0 & 0 & 0 & 0 \\ 0 & 0 & 0 & 0 \end{bmatrix},$$





and 3) using the definition of the backscattering phase matrix (Hayman and Thayer, 2012; Neely et al., 2013)

$$\bar{\bar{F}}\left(\bar{k}_i, -\bar{k}_i, R\right) = \begin{bmatrix} F_{11}\left(R\right) & F_{12}\left(R\right) & 0 & 0 \\ F_{12}\left(R\right) & F_{22}\left(R\right) & 0 & 0 \\ 0 & 0 & F_{33}\left(R\right) & F_{34}\left(R\right) \\ 0 & 0 & F_{34}\left(R\right) & F_{44}\left(R\right) \end{bmatrix} \tag{2}$$

the number of photons to be observed in any arbitrary linear polarization channel is given in Eq. 3 as

$$N_M\left(R\right) = \xi\left(R\right)\left[F_{11}\left(R\right) + \cos\left(2\theta\right)F_{12}\left(R\right) + \cos\left(2\phi\right)\left(F_{12}\left(R\right) + \cos\left(2\theta\right)F_{22}\left(R\right)\right) + \sin\left(2\theta\right)\sin\left(2\phi\right)F_{33}\left(R\right)\right]. \tag{3}$$

Here, all constant terms of Eq. 1, which will cancel when taking signal ratios, are lumped into the term $\xi\left(R\right)$ such as the

measurement solid angle, geometric overlap, range resolution, and atmospheric transmission.

The number of measured photons incident upon the photodetector, $N_M\left(R\right)$, is a function of transmitted and received polarization angle $\phi$ and $\theta$, respectively, and is related to the scattering phase matrix terms, $F_{11}\left(R\right)$, $F_{12}\left(R\right)$, $F_{22}\left(R\right)$, and $F_{33}\left(R\right)$, which are all functions of range. For CAPABL, $\phi = 45^o$; applying this constraint to Eq. 3 cancels the functional dependency on $F_{22}\left(R\right)$ by design. Thus, using three distinct receiver polarization channels: $\theta_1$, $\theta_2$, and $\theta_3$, one can create a set of three

simultaneous equations which can be inverted to calculate the Mueller matrix terms of interest that describe backscattering coefficeint ($F_{11}$), volume depolarization ($F_{33}/F_{11}$), and volume diattenuation ($F_{12}/F_{11}$). This set of equations is given in Eq. 4 as

$$\begin{bmatrix} N_1\left(R\right) \\ N_2\left(R\right) \\ N_3\left(R\right) \end{bmatrix} = \xi\left(R\right)\begin{bmatrix} 1 & \cos\left(2\theta_1\right) & \sin\left(2\theta_1\right) \\ 1 & \cos\left(2\theta_2\right) & \sin\left(2\theta_2\right) \\ 1 & \cos\left(2\theta_3\right) & \sin\left(2\theta_3\right) \end{bmatrix}\begin{bmatrix} F_{11}\left(R\right) \\ F_{12}\left(R\right) \\ F_{33}\left(R\right) \end{bmatrix} \rightarrow \bar{N} = \bar{\bar{A}}\bar{F}. \tag{4}$$

The general matrix inverse of $\bar{\bar{A}}$ is given in Eq. 5 as

$$\bar{\bar{A}}^{-1} = \frac{1}{\zeta}\begin{bmatrix} \sin\left(2\theta_2 - 2\theta_3\right) & \sin\left(2\theta_3 - 2\theta_1\right) & \sin\left(2\theta_1 - 2\theta_2\right) \\ \sin\left(2\theta_3\right) - \sin\left(2\theta_2\right) & \sin\left(2\theta_1\right) - \sin\left(2\theta_3\right) & \sin\left(2\theta_2\right) - \sin\left(2\theta_1\right) \\ \cos\left(2\theta_2\right) - \cos\left(2\theta_3\right) & \cos\left(2\theta_3\right) - \cos\left(2\theta_1\right) & \cos\left(2\theta_1\right) - \cos\left(2\theta_2\right) \end{bmatrix}. \tag{5}$$

Note that the matrix $\bar{\bar{A}}$ and the matrix inverse $\bar{\bar{A}}^{-1}$ are not functions of range but only of the selected receiver polarizations. The term

$$\zeta = \cos\left(2\theta_3\right)\left(\sin\left(2\theta_2\right) - \sin\left(2\theta_1\right)\right) + \cos\left(2\theta_1\right)\left(\sin\left(2\theta_3\right) - \sin\left(2\theta_2\right)\right) + \cos\left(2\theta_2\right)\left(\sin\left(2\theta_1\right) - \sin\left(2\theta_3\right)\right) \tag{6}$$

is introduced in Eq. 5 as a constraint on the validity of the inversion where $\zeta = 0$ results in a degenerate inversion because of

receiver polarization selection. This happens for example when two angles are equal or $180°$ separated.

Volume depolarization, hereafter referred to as depolarization,

$$d\left(R\right) - 1 = \frac{F_{33}\left(R\right)}{F_{11}\left(R\right)} = \frac{\left(\cos\left(2\theta_3\right) - \cos\left(2\theta_2\right)\right)N_1\left(R\right) + \left(\cos\left(2\theta_1\right) - \cos\left(2\theta_3\right)\right)N_2\left(R\right) + \left(\cos\left(2\theta_2\right) - \cos\left(2\theta_1\right)\right)N_3\left(R\right)}{\sin\left(2\theta_2 - 2\theta_3\right)N_1\left(R\right) + \sin\left(2\theta_3 - 2\theta_1\right)N_2\left(R\right) + \sin\left(2\theta_1 - 2\theta_2\right)N_3\left(R\right)} \tag{7}$$




and volume diattenuation, hereafter referred to as diattenuation,

$$D(R) = \frac{F_{12}(R)}{F_{11}(R)} = \frac{(\sin(2\theta_3) - \sin(2\theta_2))N_1(R) + (\sin(2\theta_1) - \sin(2\theta_3))N_2(R) + (\sin(2\theta_2) - \sin(2\theta_1))N_3(R)}{\sin(2\theta_2 - 2\theta_3)N_1(R) + \sin(2\theta_3 - 2\theta_1)N_2(R) + \sin(2\theta_1 - 2\theta_2)N_3(R)} \tag{8}$$

can be expressed in terms of arbitrary observation angles assuming the condition $\zeta \neq 0$ (for CAPABL $\zeta \approx -2$ calculated from receiver polarizations via atmospheric calibration performed for each measurement).

## 2.2 Retrieval Assumptions

By assuming the more general form of the backscattering phase matrix, Eq. 2, which allows for horizontal orientation of scatterers as opposed to only random orientation, and observing scatterers in an off-zenith direction (for CAPABL the tilt angle from zenith is $32°$), no ambiguity arises in the interpretation of depolarization measurements as seen for example by Thomas et al. (1990) or Winker et al. (2009) where low depolarization, typically associated with liquid, from ice is observed from organized specular reflections off of HOIC. Equations 7 and 8 are valid for randomly or horizontally oriented axially symmetric scatterers. If randomly orientated ice crystals (ROIC) are observed, diattenuation will be strictly $D = 0$ and the scattering Mueller matrix smplifies to a function of two elements, depolarization $d$ and the volume backscatter coefficient $\beta$. This form of the backscatting phase matrix is consistent with the works of Mishchenko and Hovenier (1995); Flynn et al. (2007); Gimmestad (2008); Hayman and Thayer (2009), and Hayman and Thayer (2012).

Traditional volume depolarization ratio, hereafter referred to as depolarization ratio, measurements are made by assuming random orientation of particles and using only two measurements of the polarization of the backscattered signal, one that is linear and parallel to the outgoing laser polarization and one that is linear and perpendicular to the outgoing laser polarization (Schotland et al., 1971; Sassen, 1991; Mishchenko and Hovenier, 1995; Gimmestad, 2008; Hayman and Thayer, 2012). Depolarization, $d$, and depolarization ratio, $\delta$, can be related but are not equivalent. Depolarization is an element of the Mueller formalism and can be measured with any set of 2 polarizations (assuming randomly oriented particles), and the depolarization ratio is often related to the phase of atmospheric scatterers but is only measured with parallel and perpendicular polarizations. They are related as

$$\delta(R) = \frac{N_{0_\perp}(R)}{N_{0_\parallel}(R)} = \frac{d(R)}{2 - d(R)} \tag{9}$$

where $N_{0_\perp}$ is the number of photons (or equivalently the photon arrival rate) at the detector surface in the perpendicular channel as a function of range, and $N_{0_\parallel}$ is the number of photons (or equivalently the photon arrival rate) at the detector surface in the parallel channel as as function of range. Measuring orthogonal polarizations imposes a stringent requirement on a lidar system that can be lessened by using the more general form given in Eq. 7.

Implicit in the development of the SVLE, and most lidar retrievals, is the assumption that the observed signal is linearly related to irradiance of light at the receiver. For targets with low depolarization ratios like liquid and clear air, the signal dynamic range in the parallel and perpendicular channels can be dramatically different. A depolarization ratio of 1% would indicate the two signals would be different by 2 orders of magnitude whereas a depolarization ratio of 50% would indicate the





two signals would be different by a factor of 2. This difference is of practical concern as most observing systems have limited dynamic range, on the order of 4 to 5 orders of magnitude.

The expressions given in Eq. 7 and Eq. 8 are generalizations of the equations presented by Neely et al. (2013) that assume fixed orthogonal receiver polarization angles. The diattenuation equations presented by Neely et al. (2013) in their Eq. 7 and Eq. 20 can be recovered from our Eq. 8 by using $\theta_1 = 45^o$, $\theta_2 = -45^o$, and $\theta_3 = 0^o$ for their Eq. 7 and $\theta_1 = 45^o$, $\theta_2 = -45^o$, and $\theta_3 = \pm 90^o$ for their Eq. 20. The depolarization term presented by Neely et al. (2013) in their Eq. 8 can be recovered with either set of angles from our Eq. 7. For clarity, retrievals performed with equations from Neely et al. (2013) are referred to

as traditional or orthogonal as the polarizations used are orthogonal in Poincare space. The retrievals using Eq. 7 and 8 are referred to as non-orthogonal as they require no such assumption.

Finally, Eqs. 7 and 8 are also derived on the strict assumption that the lidar system emits a linear polarization and measures only linear polarizations (displaying no systematic retardance for example). These assumptions have been questioned for some optical systems, e.g. Hayman and Thayer (2009) or Di et al. (2016), but have been directly measured for CAPABL. CAPABL

has a transmitter polarization purity of 123:1 and a receiver polarization purity of $> 800 : 1$, resulting in a system bias in the depolarization ratio no greater than $0.8\%$.

## 2.3   Diattenuation

Observed depolarization ratios are a function of atmospheric scattering, optical system setup, and recording systems. Traditional two-channel polarization systems can not unambiguously measure atmospheric depolarization without additional infor-

mation. Separating atmospheric depolarization from systematic effects is non-trivial. Alvarez et al. (2006) show, for example, how to calibrate differential detector sensitivity and receiver cross-talk, while Hayman and Thayer (2009) show how to remove depolarization effects caused by receiver optical retardance and scattering. However, recording systems that are subject to saturation, or underrepresentation of signal strength compared to incident irradiance, can also cause depolarization ratio effects, which are not constant in range and can not be calibrated using methods like that presented in Alvarez et al. (2006) or Hayman

and Thayer (2009). Depolarization effects related to saturation couple polarization measurements with terms in the SVLE like cloud base height, range, and optical thickness through signal intensity measurements.

The CAPABL system requires at least 3 polarization measurements to retrieve $F_{11}(R)$, $F_{12}(R)$, and $F_{33}(R)$. However, saturation has been observed to cause biases in CAPABL measurements using only 3 polarizations, i.e. inability to measure all 3 signals over the entire dynamic range leading to an underrepresentation of signal strengths and causing biases in polarization

retrievals. Thus, a fourth polarization channel is included, three to measure atmospheric properties and one to monitor recording system effects. For CAPABL, the $F_{12}(R)$ term is measured twice using two sets of polarization channels with opposite sensitivity to saturation. If the $F_{12}(R)$ terms measured in two different ways are consistent at a given altitude, the lidar counting system is operating in a linear manner. An advantage of this over-constrained polarization retrieval is that CAPABL can actively monitor if the polarization measurements are acting properly or are causing systematic biases. A combination of any 3 of the

4 polarization channels can be used to optimize CAPABL's retrievals if the polarization signals are not subject to saturation. If $F_{12}(R)$ is zero, i.e no HOIC are present or the system is insensitive to orientation (as is the case within a few degrees of zenith



or nadir), only 2 of the 4 channels are needed for atmospheric properties. However, if the polarization retrievals are subject to saturation, CAPABL's additional channels can be used to identify measurements with non-physical retrieved values and separate them from geophysical values. Therefore, including an extra polarization measurement and retrieving diattenuation can be used to verify two major assumptions: the presence of strictly randomly oriented ice crystals (ROIC) and counting system linearity.

## 3 CAPABL Hardware, Data Analysis, and Classification

The theory described in Sect. 2 is, in principle, valid for any measurement system type and polarization angle selection. However, as a practical matter, limitations in measurement systems must be considered. Measurement system sensitivity and dynamic range are the main concern for this work and, in particular, the limited observational dynamic range of signals.

Broadly, lidar counting systems can be classified as either photon counting systems or as analog systems. Photon counting systems are capable of measuring weak light signals, which allow them to observe high altitudes effectively (relative to analog

detection assuming ground based measurements). Analog systems sacrifice sensitivity to measure stronger signals, which facilitates measurement of low altitudes. In photon counting, detector signals are discriminated with a fixed voltage threshold. This threshold is set to remove much of the electrical noise resulting from using single-photon, high-gain detectors. When a voltage signal is observed in excess of the threshold, a photo-electron is counted and its time of flight is assigned to a particular time bin. The intensity is presumed to be linearly related to the total number of counts in that bin over some integration period.

Error can arise with this technique, however, if photons arrive at the counting system in close succession (Whiteman et al., 1992; Donovan et al., 1993). It is possible that pulses can pileup in such a way that two or more pulses either overlap in time or pass through the system faster than the counting system can reset itself. In either case, the intensity observed by the optical system is not linearly proportional to the number of photo-electrons counted because some photo-electrons have not been counted. In analog detection, the discrimination threshold is removed and the voltage produced by the detector is passed

through an analog-to-digital converter with its amplitude providing the relative intensity of the collected backscattered signal. This method requires much higher signal-to-noise ratio than photon counting because it lacks a discriminator that separates the influence of detector circuit electrical noise from the desired signal.

### 3.1 CAPABL Hardware

The CAPABL system has been deployed to Summit, Greenland within the ICECAPS sensor suite since 2010 (Shupe et al.,

2013; Neely et al., 2013). Since its installation several hardware modifications, completed in June 2015, have improved the system's overall observational capacity. These modifications are described with an emphasis on how they allow the CAPABL system to better observe clouds via enhancement of counting system dynamic range. The current system specifications are given in Table 1, which can be compared to Table 1 from Neely et al. (2013) for reference.

After several years of data collection, the original Nd:YLF laser was replaced by a more powerful Nd:YAG laser. This

changed the laser wavelength from $523\,nm$ to $532\,nm$. The optical components were accordingly changed. In addition, the




original 35.6 $cm$ telescope was replaced by a smaller 20.8 $cm$ Schmidt Cassegrain telescope to allow the system to be more easily tilted; the current tilt angle, set in June 2015, is $32°$ from vertical. The photo-multiplier tube (PMT) was upgraded from the original PMT, a Thorn EMI 9863B/100, to a Hamamatsu R7400U-03. These modifications have enhanced the power aperture product and the detection sensitivity of the system, which increased the overall signal-to-noise ratio.

The second major change was an upgrade of the receiver counting system from a purely photon counting system to a combined analog and photon counting system. By using a counting system that combines photon counting and analog detection, saturation in photon counting caused by high count rates is ameliorated by using analog detection, approximately $> 10\,MHz$,

while maintaining sensitivity to low count rates, approximately $< 1\,MHz$, using photon counting detection. More about this counting system can be found in Newsom et al. (2009).

CAPABL observes 4 non-orthogonal receiver polarization channels. These polarizations are all linear and were oriented parallel to the outgoing polarization, $0°$, perpendicular to the outgoing polarization, $90°$, approximately $30°$ from parallel (or $60°$ from perpendicular) polarization (referred to as 3rd channel), and approximately $110°$ from parallel (or $20°$ from

perpendicular) polarization (referred to as 4th channel). Combining the new counting system and the 4 linear polarizations with the non-orthogonal polarization theory, this allows 8 methods to invert and solve Eq. 4.

## 3.2   Processing Methods

Data analysis and classification is performed by taking advantage of CAPABL's variety of polarization signal measurements. There are several levels of processing and filtering to ensure data quality. These are implemented in an automatic algorithm.

The steps are given in Table 2 and described here in order.

CAPABL makes observations with 5 sec resolution per polarization angle and scans through 4 polarization angles before returning to the original polarization, taking a total of 20 sec before returning to the first polarization angle. The outgoing polarization is $45°$. These scans are parsed by like polarizations and time integrated to 20 sec per polarization and spatially integrated to the resolution of 30 m. Non-paralyzable saturation corrections are applied per the method described in Appendix

A and by Whiteman (2003) to the photon counting data. Note that the variance of saturation-corrected photon counting is not simply the variance from Poisson statistics, but also the error introduced by an inexact model fit is taken into account for all error analyses and is described in Appendix A. All data is then background subtracted and subject to an SNR filter. Photon counting data with less than one photon count per bin after background subtraction and analog voltages less than 1 mV per bin after background subtraction (SNR ratio of approximately -8 dB) are removed. This background subtracted and SNR filtered

data is then passed through a speckle filter, which interrogates a 5 by 5 time and altitude bin region, referred to here as a voxel, around each voxel of interest. Voxels, where more than 75% of the surrounding data are removed by the SNR filter, are also removed.

Depolarization, depolarization ratio, and diattenuation as well as their error estimates are calculated using the orthogonal polarization approach presented by Neely et al. (2013), and also using the non-orthogonal approach described here. The or-

thogonal approach uses all the same steps as Neely et al. (2013) but with the following exception. Instead of assuming the observations are made at exactly 1) parallel, $0°$, 2) perpendicular, $90°$, and 3) $45°$, the angle of the third channel is carried



through the analysis as a variable and the retrieved angle from atmospheric calibration is used. This is designed to accommodate for slight retardance changes in the liquid crystal variable retarder (LCVR) as a function of ambient laboratory temperature. For the depolarization retrieval in areas that lack oriented scatterers, the depolarization can be calculated with any two of the receiver polarization channels. HOIC are identified by non-zero diattenuation, $D$. Diattenuation is calculated in two ways, 1) using parallel, perpendicular, and the 3rd channel referred to as $D_1$ and 2) using parallel, perpendicular, and the 4th channel referred to as $D_2$. These channels are chosen because of their opposite sensitivity to saturation for the photon counting and saturation corrected photon counting retrievals. By multiplying the two measurements together, negative values indicate $D_1$

and $D_2$ are tending in opposite directions indicating a saturation event. Conversely, positive values of $D_1 D_2$ indicate the two measurements are tending together and that the non-zero diattenuation is physical, i.e. unaffected by saturation.

Data is removed outside of the allowable ranges: $0 \le d \le 1$, $0 \le \sigma_d \le 0.4$, $-1 \le D \le 1$, and $0 \le \sigma_D \le 0.2$, as these represent non-physical conditions. Note that the error analysis procedure for photon counting described by Neely et al. (2013) assumes Poisson statistics where the data is assumed shot noise limited. The same procedure for photon counting is carried

through the analysis shown here. The analog signal is not governed by Poisson statistics however. The analog uses the variance of the background voltages for its error estimates. Additionally, as mentioned above, the variance for saturation corrected photon counting is modified to reflect the correction procedure and the variance introduced via inexact model fitting. Finally the backscattering ratio, the ratio of total scattering to molecular scattering, is calculated. Expected molecular scattering is calculated using temperature and pressure information collected from the ICECAPS twice daily radiosonde program, interpolating

between launches. The inversion technique of Klett (1981) is used to calculate total scattering coefficient as described by Neely et al. (2013). A lidar ratio of 10 is assumed, following the review of Nott and Duck (2011) and references therein, to convert the total extinction derived by the Klett inversion to total backscattering coefficient.

By design, CAPABL uses 4 polarization channels to measure 3 elements of the scattering Mueller matrix: $F_{11}$, $F_{12}$, and $F_{33}$ with one additional measurement to monitor saturation. If saturation is not an issue, any 3 of the 4 channels may be used for

the inversion of polarization properties. Thus, the utility of the generalization given in Sect. 2 is that the 3 signals with the least error can be used at any time. For example, the 3 strongest signals for measurements of high ice clouds where backscattered signals are weaker or the 3 weakest measurements for low liquid clouds where the backscattered signal is stronger. Using non-orthogonal polarizations allows the dynamic range between polarization components to be accommodated and optimized.

### 3.3 Classification

Using all of the polarization processing listed above, the classification of data is performed in the following manner. Clear air is found as any voxel with a backscattering ratio less than 2.6. Sub-visible clouds and aerosols are any voxel with a backscattering ratio between 2.6 and 6.5. Clouds are tagged as voxels with backscattering ratio greater than 6.5. For reference, Cesana and Chepfer (2013) use a threshold value for backscattering ratio of 5 to identify cloudy scenes. Within cloud voxels, the depolarization ratio threshold, originally defined by Intrieri et al. (2002) of $\delta_O \ge 0.11$ was used to define ice and $\delta_O < 0.11$

as water. Any voxels tagged as aerosol that displays a depolarization ratio $\delta_O \ge 0.11$ is reset as ice. HOIC are identified by $D_1 D_2 > 0.01$ with $\sigma_{D_1}, \sigma_{D_2} \le 0.05$.



Classified lidar profiles can then be condensed to a single column classification for the radiatively dominate species, referred to as the column type. If a column contains liquid voxels at any altitude, the column is labeled as liquid. If a column lacks liquid but contains ice voxels, it is labeled as ice. Ice is separated into 2 categories. If the column is labeled ice and contains HOIC at any altitude, it is labeled HOIC otherwise ROIC. If the column contains no liquid or ice but contains sub-visible voxels, it is labeled sub-visible. Finally, if the column lacks all other types of voxels, it is labeled as clear. One note is that this method can misclassify areas that lack lidar data as clear air. Lidar data can be lacking due to attenuation due to low-lying fog, clouds below lidar overlap, or an obstructed lidar window. In this case, data can be mistakenly classified as clear air. As a final check,

data that is classified as clear air must have substantial signal above 2 km altitude otherwise it is tagged as obscured instead of clear air.

The thresholds set for the automated classification algorithm are important to the interpretation of the results of this work. Depolarization and diattenuation are both elements of Mueller matrices, which are defined to have absolute values less than or equal to unity. Values outside this are non-physical. The values on the depolarization and diattenuation error bounds are limited

mostly by background irradiance, which is tuned via receiver hardware. A receiver neutral density filter lowers both the signal count rates and atmospheric background count rate by a factor of 1000, which brings the majority of the signal intensity into the desired dynamic range of the counting system and makes the depolarization and diattenuation error values limited only by shot noise. The filters, which remove data points based on depolarization and diattenuation and their errors, remove less than 3% of all data values. For context, background and speckle filters remove approximately 60% and 23%, respectively, of all

data points.

The setting of the backscatter ratio bounds is more subjective. As there is no true molecular measurement at Summit (for example provided by a Raman lidar or high spectral resolution lidar), the Klett inversion was used assuming a lidar ratio of 10. Curry et al. (1996) note that clear air is uncommon in the Arctic. It has been the authors' experience that even the clearest days at Summit still have some amount of ice in the sky. The clearest day observed within May and June 2015 is used as a

baseline to set the clear air threshold of 2.6. The threshold limits between aerosol or sub-visible clouds and clouds were set using an all sky camera. The thinnest visible cloud layer observed during the same time period was used to separate the aerosol or sub-visible clouds and cloud classifications.

The threshold between liquid and ice, $\delta_O = 0.11$, is taken from literature related to the Depolarization and Backscatter Unattended Lidar (DABUL) (Intrieri et al., 2002; Shupe and Intrieri, 2004; Zuidema et al., 2005; Shupe et al., 2006), which

was the predecessor to CAPABL, and not changed for this work. An analysis was performed (not shown) of the effect of this threshold on cloud fractional occurrence (FO), the ratio of a particular single column classification type to all measurements. This analysis shows that thresholds between $\delta_O = 0.11$ and $\delta_O = 0.2$ change the FO of liquid and ice by less than 1% over the period of a month for July 2015. Thresholds below $\delta_O = 0.11$ significantly alter the FO of liquid and ice making $\delta_O = 0.11$ a reasonable threshold value.



## 3.4 Algorithm Examples

An example of this data classification procedure is given in Fig. 1 for analog detection and Fig. 2 for photon counting detection for February 29, 2016. This day is chosen because it contains both single level and two level mixed-phase cloud systems as well as high ice clouds. In comparing these two figures in the first 12 hours of the day, the mixed-phase cloud layer at approximately 1.5 km altitude has been identified with substantially more liquid voxels when classified using analog detection than using photon counting detection. Furthermore, there are two smaller mixed-phase cloud layers that exist below 1 km between 3 and 5 UTC and 8 to 11 UTC identified by analog detection, which are interpreted as purely ice when classified with photon counting observations. This discrepancy in interpretation is directly linked to cloud macrophysical properties, such as base height and optical depth that result in high count rates and cause saturation of the photon counting parallel channel. This

increases the observed depolarization ratio by reducing the parallel photon count rate beyond the liquid-ice threshold and alters the derivative of the signal intensity that affects the Klett inversion.

To demonstrate that the day selected is not anomalous, monthly statistics are compiled for the first 4 months of data available, July 2, 2015 to October 31, 2015, since the hardware updates described. Over this time, the CAPABL system ran continuously and had an uptime of $> 99\%$ (this equates to approximately 5 minutes of missed data per day, which occurs at midnight UTC

each day to perform system diagnostics and housekeeping). Voxels are separated by cloud phase and clear air. Voxels are integrated over a month-long period for each altitude and time bin. These data are compiled into box-and-whisker plots given in Fig. 3. The median altitude of all voxels for each identifier: ice, liquid, and clear air, is given as a line through the center of the box, which is completed by the 25th and 75 percentile of all monthly data. The whiskers extend to the 5th and 95th percentiles. The other data values are considered outliers.

Figure 3 indicates 3 prominent features. First, the median altitude of liquid voxels is not constant between analog, photon counting, and saturation corrected photon counting (SCPC) for either orthogonal or non-orthogonal retrievals. There is a clear 1 to 2 km offset in the medians between analog and photon counting (1.72 km, 1.43 km, 0.75 km, and 0.91 km offsets for July, August, September, and October, respectively). This offset in mean voxel height indicates that low-level, liquid clouds are often misclassified by the photon counting channel as indicated by Fig. 1 and Fig. 2. The second feature is seen in the clear

sky data where there is increased sensitivity of the photon counting channel over the analog channel and increased sensitivity of the non-orthogonal polarization retrievals over the orthogonal versions. This increased sensitivity is seen by the increase in whisker range of approximately 1 km (0.96 km, 0.70 km, 0.34 km, and 0.55 km for July, August, September, and October for saturation corrected photon counting and analog to the 95th percentile, respectively, or 1.17 km, 1.12 km, 0.99 km, and 0.83 km to the inner fence) indicating the presence of more high altitude clear air voxels that pass the quality control standards

specified in Table 2. As a result of the increased sensitivity, the median altitude of the clear-sky data shifts upwards as well (0.29 km, 0.29 km, 0.36 km, and 0.31 km for July, August, September, and October for SCPC, respectively). The final feature is the relative consistency of the occurrence of ice for all methods. The median altitude of the ice-identified data shifts slightly upwards again due to increased sensitivity between analog and photon counting (0.05 km, 0.23 km, 0.36 km, and 0.23 km for July, August, September, and October for saturation corrected photon counting and analog, respectively) but the boxes





cover similar altitude ranges, especially for July. Comparing the whiskers for the non-orthogonal and orthogonal polarization retrievals within a month indicates that the increased sensitivity gained by using non-orthogonal polarization retrievals does not change the geophysical interpretation of the ice-identified data when saturation is of little concern (shifts of 0.26 km, 0.08 km, 0.21 km, and 0.10 km for July, August, September, and October for analog to the 95th percentile, respectively, or 0.18 km, 0.13 km, 0.21 km, and 0.18 km to the inner fence are observed), i.e. when signals are of similar strength or when signal rates are less than or on the order of approximately 1 MHz.

bs

## 4 Merged Best Estimate Cloud Product

The classification results of Fig. 1, 2, 3 raise the question, "What retrieval technique is most accurate?". None of the results presented is perfect as each technique has innate benefits based on counting system dynamic range. For example, analog detection is designed for stronger signals, and photon counting detection for weaker signals. A single combination of all of the CAPABL data products leverages all of the advantages of analog and photon counting observations as well as non-orthogonal polarization retrievals to extend the dynamic range of the counting system. This section describes the broad rules used to combine all of the possible data collected into a single best estimate profile. This merging is done on the basis of signal counting regimes. Here valid signal ranges are defined where the measured signal count rate is linearly proportional to incident intensity at the detector. For analog detection, the range is fixed by the analog noise in the detector circuit on the low end and by the width of the analog-to-digital converter (ADC) bounds on the high end. For photon counting, the range is fixed by the discriminator threshold and pulse height distribution on the low end and detector and counting system dead time on the high end.

The SNR filter and the speckle filter are designed to remove data lacking signal strength in one or more of the polarization signals. These filters are applied to all data streams individually (to each polarization and counting type) and provide a lower limit of acceptable count rates for all channels. This limit is much higher for analog detection (approximately 1 MHz) and much lower for photon counting detection (approximately 10-100 kHz). The upper limit of count rate is enforced via bounds set on the receiver ADC. The analog counting system is able to track PMT signals that exceed the ADC bounds. This occurs either with a PMT pulse that is too large or with multiple PMT pulses piling up in succession or with a pulse that has too large of a voltage rebound. The ADC bounds are set from -495 mV to 5 mV with negative tending detector signals, which are nominally set to result in PMT pulses of approximately 10-15 mV. In all cases, if any shot results in any altitude bin signal on any polarization outside the valid ADC range, that altitude bin is removed from the data stream (hereafter referred to as clipping). Such clipped signals compose approximately 0.78% of all data from 0 km to 8 km and are removed from both analog and photon counting detection data streams as they represent counting data that are no longer linearly proportional to incident intensity.

Applying the above filters to analog and photon counting raw data forces the data outside the valid counting range to be removed. For the analog signal, the data above the valid counting range is removed by the clipping filter, and the data below





the valid count range is removed by the SNR and speckle filters. For the photon counting signal, the data below the valid count range is also removed by the SNR and speckle filters. The upper range of photon counting signal is however not necessarily limited by the clipping filter. In fact it is still poorly constrained due to possible pulse pileup. To specify the upper bound of the valid signal range for photon counting signals, the combination of analog and photon counting is considered. Implicit in the combined detection of analog and photon counting data is the assumption that there exists a range of counting signals, in the range of approximately 1- 10 MHz, where both signals are acting linearly, i.e. that both measurement values reported are linearly proportional to the incident intensity at the detector. By this assumption, all data measured by the analog channel will be an upper bound on the photon counting detection. Practically speaking, this means that data removed from the analog detection scheme by the SNR and speckle filters is potentially valid photon counting data. Saturation corrected photon counting is not needed.

All data types are processed as described in Sect. 3 removing all invalid signals. Data is stitched together by first taking all valid orthogonal analog signals. Any locations where valid orthogonal photon counting signals are present that are not previously covered by analog are then added. Non-orthogonal data using the 3 strongest signals for analog first then photon counting are then added where available. Non-orthogonal data using the 3 weakest signals for analog only is then added to fill low altitude areas that may have been removed due to the parallel channel's clipping filter.

There exists another way of viewing analog, photon counting, and saturation corrected photon counting data, which is presented by Newsom et al. (2009), referred to as gluing. This work will not perform the gluing procedure presented for several reasons: first it is impractical to calculate gluing coefficients by atmospheric calibration as access to the CAPABL system is limited to once or twice a year, second it is not clear how to combine analog and photon counting signals at a single height to adequately account for error introduced by temporal variation of gluing coefficients, and third it is not clear how the range correlation of signals required for the Klett inversion method is affected by the thresholds of the gluing procedure.

In contrast to these issues with data gluing, the method described above and used for this work addresses these problems in the following ways. Primarily, there is no need to track the temporal variation of gluing coefficients. By performing polarization retrievals as described, the time dependence of the detector is effectively canceled by ratio values of the polarization measurements. This method effectively reduces the assumption of a time variance in the detector from hours to the time it takes to make a complete polarization measurement set, which for CAPABL is 20 sec. Additionally, the range correlation required by the Klett inversion is preserved by considering each type of profile individually. Moreover, by systematically verifying each detector signal is within the counting system's observable and valid dynamic range, polarization retrievals can track Poisson or Gaussian errors (associated with photon counting and analog detection, respectively) in a more accountable way. Finally, as a practical matter, access to CAPABL occurs approximately once or twice per year. The method used allows the optical attenuation in the receiver to be set once and left untouched for the year.

An example of the merged data product is given in Figure 4 for August 22, 2016. The raw analog signals are provided in the top panel, the merged data product in the middle panel, and the origin of the data for each pixel in the lower panel. This procedure takes most of its data from analog detection during daytime and low cloud scenes, much more data from photon counting during nighttime, and in the upper clear air and cloud scenes from non-orthogonal retrievals.





Considering the 4 month period of Fig. 3, monthly FO values are calculated by CAPABL from its column data classification. FO is calculated for all types of data processing as well as the best estimate data product in Fig. 5. Figure 5 clearly illustrates several features. First, photon counting and saturation corrected photon counting dramatically underestimate the occurrence of liquid clouds. This is because both methods have strong saturation induced biases, linked to cloud base height and optical depth, which lower the observed parallel count rate artificially raising the observed depolarization and consequentially depolarization ratio. This serves to flip the classification of most water clouds to ice clouds. Second, analog detection underestimates overall cloud fraction due to its sensitivity, i.e. analog detection only sees clouds that are lower and more optically thick but misses many high tenuous clouds. Finally, in all cases, the merged data has less clear air than the single measurement techniques caused by an extended dynamic range and altitude range of observable signals.

## 5 Multisensor Validation

Validation of remote sensing instrumentation that lack traceable calibration standards such as polarization lidars is of particular importance (Freudenthaler, 2016). This section evaluates the CAPABL cloud identification data product by using ancillary measurements taken by the ICECAPS program, namely a co-located micro-pulse lidar (MPL), millimeter cloud radar (MMCR), microwave radiometer (MWR) and broadband radiation measurements. The period of comparison is from July to December 2016. For this period, each sensor had an uptime of better than 95%; one major reason for the period selected is the simultaneous measurement of much of the ICECAPS suite. This period also covers both polar day and night.

One important note for interpreting the results presented is the instrument pointing angle for CAPABL, MPL, MWR, MMCR, and radiation measurements. CAPABL operates at $32°$ off zenith, the MPL operates at approximately $5°$ off zenith, and the MMCR within $0.2°$ of zenith. The radiation measurements are approximately $600\,\mathrm{m}$ away from CAPABL, MPL, MWR, and MMCR measurements and are total hemispheric measurements instead of narrow field of view. Given these constraints, the assumption of horizontal homogeneity of the scene above the site on the order of $500\,\mathrm{m}$ is required for an average voxel height of $2\,\mathrm{km}$.

### 5.1 Colocated Instruments

#### Micro-Pulse Lidar (MPL)

The MPL used in this work is a Sigma Space V4 polarization sensitive system provided to the project by the ARM Program. The MPL uses a frequency doubled Nd:YAG laser at $532\,\mathrm{nm}$. The system hardware design is well described by Campbell et al. (2002) and the polarization hardware and retrievals by Flynn et al. (2007).

MPL data is processed as follows. MPL raw data (photon counts) are time and space integrated as close as possible to CAPABL's data grid. Calibrations as described by Campbell et al. (2002) are performed monthly to remove signal induced noise (SIN) resulting from the strong light signals from the shared telescope transceiver design. The SIN calibration corrections applied are linear interpolations between subsequent SIN calibrations. The calibration data is taken at $30\,\mathrm{m}$ resolution, which





sets the lidar range resolution of this study. This SIN corrected raw data is then linearly interpolated from the MPL grid directly to the CAPABL grid. The polarization properties are calculated as in Flynn et al. (2007) with no modification to the method presented. Note that the MPL measures depolarization using both linear and circular polarizations while CAPABL measures only linear polarizations. A comparison of the specifications of CAPABL and the MPL is given in Table 3.

MPL data is classified as for CAPABL. Note that the MPL is only a photon counting system while CAPABL uses both analog and photon counting, and CAPABL has the unique ability to measure the $F_{12}$ element of the scattering matrix upon which the diattenuation measurement is based. Filtering steps based on diattenuation and classification for HOIC are not performed by the MPL given the inability to make $F_{12}$ measurements. For this study, MPL data results in voxel classifications that are either clear air, cloud ice, cloud liquid, or removed due to data filtering.

**Millimeter Cloud Radar (MMCR)**

The MMCR used in this study was developed and provided by the National Oceanic and Atmospheric Administration's (NOAA) Earth Systems Research Laboratory. The MMCR is a 35 GHz single polarization Doppler radar. A general hardware description is given by Moran et al. (1998) and its software and operational measurement modes documented by Clothiaux et al. (1999). Data products available are based on observed Doppler spectra. Specifically, the system reports reflectivity (the integral of power in the Doppler spectrum), mean Doppler velocity (the first moment of the Doppler spectrum), and Doppler spectral width (the second moment of the Doppler spectrum). The zenith-pointing system occupies space in the same building as CAPABL and is carefully leveled by an onsite technician as needed to within approximately 0.2° of zenith as the snow on which the building sits settles.

Data used for this study are from the radar general mode and high sensitivity mode, referred to here as cirrus mode, with some operational settings given in Table 4. Radar data is generally taken at higher temporal resolution and lower spatial resolution than CAPABL. To push the radar data onto a similar grid as CAPABL and the MPL, radar data is incoherently averaged in time to as close to the CAPABL grid as possible. Then, as with the MPL, data is linearly interpolated in time and space to the CAPABL grid.

**Microwave Radiometer (MWR)**

Column moisture measurements are calculated using two co-located MWRs manufactured by Radiometer Physics GmbH (RPG). The first radiometer, an RPG Humidity and Temperature Profiler (HATPRO), samples 14 channels from 22.2 GHz to 60 GHz of which 23.8 GHz and 31.4 GHz are used to retrieve precipitable water vapor and cloud liquid water while the second radiometer, an RPG LWP-90-150, samples at 90 GHz and 150 GHz. From microwave brightness temperature observations, the column liquid water path (LWP) is retrieved using physical retrievals and an optimal estimation algorithm (Cadeddu et al., 2013). The LWP uncertainty using the 23.8 GHz, 31.4 GHz, 90 GHz, and 150 GHz data in the retrieval is approximately 5 g/m$^2$. Similar steps, incoherent averaging in time then linear interpolation, are performed as with the radar to push MWR data onto CAPABL's grid. MWR data is a column measurement so averaging and interpolation are only performed in time and are compared to CAPABL's column data product.



**Radiation**

Surface broadband radiation measurements are made at Summit by a pair of heated aspirated Kipp and Zonen CM22 pyranometers with spectral sensitivity from 0.2 μm to 3.6 μm and a pair of aspirated Eppley Precision Infrared Radiometers (PIR) pyrgeometers, sensitive to the spectral range from 3.5 μm to 50 μm. These instruments were originally installed in August

2013 by NOAA's Global Monitoring Division. The instruments are maintained by an onsite technician at Summit, including daily removal of accumulated ice or snow. Raw data is reported as 1 min averages.

    The pyranometers are calibrated every 2 years at NOAA's Solar Radiation Calibration Facility. The raw data are quality controlled by NOAA's Global Monitoring Division Radiation Group. A dome correction factor for the longwave PIR is applied similar to that of Albrecht and Cox (1977). More information about the available radiation measurements at Summit is given

by Miller et al. (2015).

## 5.2    Direct Lidar Comparisons

The first comparison performed is between CAPABL and the MPL. This is the simplest comparison to make because the data products of the MPL and CAPABL are very similar and both systems use the same operational principles. Because both instruments are lidars and have similar data streams, the results can be compared directly. As such, CAPABL's merged best

estimate voxel identifications are compared directly to the MPL's voxel identifications. Voxel identifications from CAPABL and the MPL compared for three separate time periods: July 2016, December 2016, and July-December 2016. These data are given in a confusion matrix, a classification model to compare two sets of results, in Table 5, where ROIC and HOIC voxels are both combined for this comparison into "CAPABL Ice".

    The time periods given in Table 5 are selected due to the solar background conditions. During the summer, July, the sun

is always above the horizon at Summit. During the winter, December, the sun is always below the horizon at Summit. These two cases are highlighted to show the difference solar background makes on the data and in particular the effect on the MPL signals, which are affected strongly by solar background. CAPABL is less affected by solar background because of the receiver attenuation.

    Table 5 highlights some significant sensitivity improvements of CAPABL's merged data product for daytime operations

compared to the MPL. In the clear column for example, in approximately 98% of the time that CAPABL observes clear air, the MPL either agrees or lacks data to refute the CAPABL measurements over the entire study period (seen in Table 5 summing similar time periods in cells A and M). This increases to 99.5% for daylight measurements. Likewise, 96% of the data in daylight that fails the CAPABL filtering process also fails the MPL's filtering process (first line of cell P) indicating a limit of penetrable optical depth for a given power-aperture product that is a theoretical limit of all lidars. In many cases, highlighted in

boxes B and C in Table 5, the MPL observes clear air while CAPABL observed clouds of some sort. This is linked directly to the Klett inversion technique requiring a strong signal derivative to highlight large backscattering ratios, approximately $> 5.0$. In the case of many high clouds, the signal derivative is not strong due to noise in the perpendicular observation channel of the MPL. In comparison, the values highlighted in boxes E and I in Table 5 are more than two orders of magnitude smaller



because the strength of the perpendicular signal does not limit the detection range for CAPABL as it does for the MPL due to
CAPABL's non-orthogonal polarization retrievals.

The sensitivity of CAPABL is linked directly to the use of analog detection and non-orthogonal polarizations. A limitation of traditional orthogonal polarization retrievals for lidar is the fact that one channel is often of higher signal strength than the other. For the MPL, the circular polarization channel is much stronger than the perpendicular polarization for low depolarization targets like clear air and liquid water. As a result, the dynamic range of the system is partially reduced by the measurement
setup. For example, a depolarization of 1% would yield a difference in signal ranges of 2 orders of magnitude at a constant altitude. Therefore, for the system to observe such low depolarization, the system necessarily must sacrifice range. In terms of altitude, the lowest possible observations are set by the circular channel overlap considerations and the counting system dead time, and the highest possible observations are set by the SNR of the perpendicular signal. In contrast, CAPABL's minimum range is set by the second weakest of its 4 polarizations (the 3rd channel) and maximum range is set by the second strongest
of its 4 polarizations (the 4th channel). By design, the 3rd channel is approximately half of the parallel channel's intensity and the 4th channel exceeds the perpendicular signal intensity by more than an order of magnitude enhancing the observable range of the system in both high and low altitudes simultaneously. As a result, CAPABL is much more sensitive to a wider range of clouds and cloud properties because it is less constrained by its observable dynamic range.

The data presented in Table 5 for December observations show a major jump where CAPABL data fails QC filtering but
MPL data shows clear air (seen in Table 5 cell D). The filtering performed after SNR and speckle filtering by CAPABL is mostly done via the unique diattenuation measurement and diattenuation error bounds. As a result, the depolarization filters are set fairly wide as they are practically unneeded. However, for the MPL, the same bounds for the filter do not tag similar low SNR cases. As a result, CAPABL data is filtered more conservatively than the MPL given the same filtering bounds on depolarization based on the diattenuation filter that can not be applied to the MPL.

The MPL and CAPABL rarely miss detecting cloud cases when they are observable by lidar. For each background condition and for the entire length of the study, not more than 3% of data is missed by one instrument when the other instrument sees cloud activity, indicated by the maximum value in boxes H, L, N and O. However, the MPL frequently mischaracterizes clouds as clear, as highlighted in cells B and C in Table 5. This is attributed as above to the signal in high background cases being hard to determine and the Klett inversion often misses thin cloud layers.

## 5.3 Comparisons with Non-Lidar Data Sources

Comparisons of CAPABL data to ancillary, non-lidar, instrumentation is less straightforward than the comparison presented with the MPL. Instead of a direct comparison such as presented in Table 5, arguments about data consistency must be made. For example, within a mixed-phase cloud, both phases of water will have large size parameters (the radius of the particle, $r$ relative to the wavelength, $\lambda$, given as $2\pi r/\lambda$), likely greater than 50-100 when observed by lidar, whereas at the radar wavelength
the size parameter is much less than 1. In this regime, the lidar will see a scatterer well into the resonant and geometric optics regime of elastic scattering whereas the radar will see a Rayleigh scattering target. As such, the two systems respond to different aspects of the hydrometeor population; this is one major benefit for having multiple sensors.



The expectations of multi-sensor comparisons are as follows. At 35 GHz, the MMCR signal is nominally proportional to hydrometeor size to the 6th power and is thus dominated by ice because liquid water drops are much smaller in diameter than ice in the Arctic (Turner, 2005; Shupe et al., 2006, 2013). Liquid water droplets are on the order of $10^{-5}$ m while ice crystals are on the order of $10^{-4}$ m to $10^{-3}$ m. By extension, the MMCR is more sensitive to liquid water droplets than clear air. One expects therefore, to see higher radar reflectivity for ice than liquid and less still for clear air. Furthermore, as ice is much larger at Summit than liquid water droplets one expects to see higher mean Doppler velocities for ice as liquid water drops are too

small to be effectively precipitated (Morrison et al., 2012). For comparison of CAPABL's data to column measurements of liquid water path with the MWR, one expects CAPABL to identify liquid overhead while the MWR observes a positive liquid water path. Likewise, one expects to have little to no liquid water path measured for ice or clear air columns.

In light of these expectations based on observed geophysical properties, the multi-sensor comparisons are performed as follows. MWR data are processed and interpolated to the CAPABL time grid. CAPABL data are then collapsed to a column

measurement based on the most radiatively important voxel type. The MWR LWP data are then assigned to one of the 4 possible column types: clear, ice (with HOIC labeled as HOIC or without labeled as ROIC), or liquid defined by CAPABL. The probability density function of the MWR LWP characteristics are calculated from all available data for each classification type. The cumulative distribution function is then calculated and presented in Figure 6. The data that has been filtered by lidar are removed. Raw radar data that has been pushed to CAPABL's data grid is assigned using CAPABL's data identifiers. The

5 data types (clear air, cloud liquid, ROIC, HOIC, and filtered) are each distinctly binned together. The probability density function of the radar characteristics and the cumulative distribution function are also calculated from all available data. The data that has been filtered by lidar are removed. The cumulative distribution functions of the remaining 4 variables are shown in Figure 6 for the first 2 radar Doppler moments and its SNR. In this time period, CAPABL has data available for 75.3% of the total voxels where there is filtered MMCR data available. Note that though they contain and represent the same data, this

work will choose to represent instrument comparisons in terms of their cumulative distribution functions as opposed to the probability density function. Both facilitate comparisons of large quantities of data but cumulative distribution functions allow simple comparisons of differences of shape and median whereas the probability density function allows for investigations of modes and biases.

It can be seen in Figure 6 that the expected relationships between the lidar, MWR, and MMCR hold very well. Nearly 69%

of all columns tagged as containing liquid by CAPABL have non-zero LWP (here zero and non-zero are taken below and above the error bounds of the measurement, respectively). Almost 91% of columns tagged as ROIC, 90% tagged with HOIC, and 91% tagged as clear do not have LWPs above the error bounds of the MWR measurement. CAPABL can mis-identify very low cloud and precipitation, below approximately 200 m, as clear air columns because there is no identifiable cloud voxels in the instrument's valid sample volume. In terms of comparison to radar, this is not a problem as no mask is returned and is thus

not considered, but in terms of column measurements this will yield an error in identification. The values given in Fig. 6 are filtered conservatively. Further filtering increase the percent of clear air, ROIC, and HOIC data with zero LWP on the order of approximately 5%.





The reflectivity of clear air voxels is much lower than that of ice and liquid water. More than 89% of all voxels identified by CAPABL as clear fall below -20 dBZ whereas only 42% identified as ice fall below the same threshold. This is confirmed with

radar SNR where 69% of all clear air data falls below the SNR threshold of -20 dBZ (this value is 72% for the threshold of -14 dBZ used by Shupe et al. (2013)). Similarly, the largest scatterers, ROIC and HOIC, have higher SNR. Note that HOIC have a lower median reflectivity than ROIC in Figure 6. This is not true in the more sensitive radar cirrus mode above 3 km (the altitude above which radar data artifacts are not expected). The cumulative distributions for the radar cirrus mode (not shown) have reflectivity values for ROIC and HOIC that nearly overlap. This change in reflectivity and inconsistency between radar

modes could indicate two things: first that HOIC are possibly occurring in thinner more tenuous clouds on average than ROIC with smaller ice particles, or second that ground based lidar measurements have a sampling bias that only allows observations of HOIC in thinner clouds.

ROIC has the highest Doppler velocity, with HOIC and liquid falling slower. ROIC has a median mean Doppler velocity of approximately 0.57 m/s downward, while HOIC and liquid are 0.47 m/s and 0.38 m/s, respectively, both in the downward

direction. The occurrence of falling liquid indicates mixed phase voxels where CAPABL is more sensitive to the liquid phase and MMCR to the ice phase. The slight skewness of the clear air identifier to downward mean Doppler velocity, indicated by the non-zero median, indicates that some ice is being tagged as clear air by CAPABL, which is known to occur at the very top of clouds and below very optically thick clouds due to the Klett inversion, and is especially prominent as mentioned with the MWR results where low (below approximately 100-200 m) thick clouds are observed. The reduced Doppler velocity of

HOIC is anticipated due to the enhanced aerodynamic drag associated with their orientation (Westbrook et al., 2010). This is a clear verification that HOIC identification by CAPABL based on the novel diattenuation technique of Neely et al. (2013) is physically consistent.

# 6   Discussion and Conclusion

## 6.1   Applicability to Other Lidar Sensors

Polarimetric lidar systems are widely deployed. Nott and Duck (2011) and references therein lists many other ground based lidar deployment sites in the polar regions such as Syowa, Antarctica, South Pole, Antarctica, Eureka, Canada, and Barrow, Alaska. Further, the CALIOP lidar on board the CALIPSO satellite uses analog detection, and regularly observes the polar regions (Winker et al., 2009). Due to varying configurations and approaches by other lidars, it is difficult to specifically identify how well other comparable systems represent cloud properties, but the unique instrument suite at Summit and the novel lidar

configuration of the CAPABL system enable such an analysis.

Lidar systems are fundamentally limited by their receiver dynamic range. For polarized systems, like CAPABL and the MPL, observational range is inversely related to atmospheric depolarization. Assuming a limited dynamic range of 5 orders of magnitude, this can be parsed either for range, arising from the $A/R^2$ term in the lidar equation, or by depolarization. For depolarization ratios of 1%, this leaves only 3 orders of magnitude for changes in range. The altitude range is limited on top by

weak perpendicular signals and on the bottom by strong parallel signals. Fundamentally, this limits the effective observational



range that has the effect of biasing attribution of cloud properties on, for example, evaluating the radiation budget. At Summit, CAPABL provides a fully merged data product that covered 34% of the column from 0 km to 8 km for July to December 2016. In comparison to CAPABL, the MPL observed 19% of the column above Summit in summer (CAPABL observes 25%) and 44% in winter (CAPABL observes 45%). The data is split again noting that CAPABL is more conservatively filtered in the winter based on its diattenuation filtering. Thus, the general impact of lidar observations is site and lidar specific (as is analyzed for Summit in Section 6.2) but should be recognized as a possible cause for interpretive bias.

Potential shortcomings of limited counting system dynamic range are clearly visible in the data shown in this work. Figure 5 shows the FO of clouds above Summit using analog detection and photon counting detection, as well as orthogonal and
non-orthogonal polarization retrievals. Each of these observational methods can handle slightly different altitude ranges based on the signal strength and system sensitivity to those signals. The results indicate that the occurrence of liquid water can be underestimated by as much as 30% depending on the counting type. This limitation is due to low level clouds causing saturation in photon counting detection, especially in the stronger polarization channels, which overestimates depolarization, and consequently the depolarization ratio, which makes liquid clouds look like ice clouds. Photon counting systems, such as
the polarization sensitive MPL, are susceptible to this sort of underestimation of liquid water clouds. In the opposite direction, analog detection underestimates total cloud FO, on the order of 4% to 22%, because it is insensitive to higher, optically thinner, ice clouds that are clearly visible using photon counting detection. In either case, the choice of counting system type, or indeed receiver polarization selection, limits the altitude range of interest and by extension the clouds to be observed. The unique configuration of CAPABL allows for these assessments to be made and optimized.

Another clear limitation of lidar sensors is their inability to observe the entire vertical column in the presence of optically thick clouds with visible optical depths on the order of 3 or greater. This limitation is clearly visible in CAPABL's data and in particular its incomplete coverage of the entire altitude range above Summit during times of mixed-phase clouds. Similar limitations are to be expected from both ground-based systems and space-based systems. An analysis of ground- and space-based observations of HOIC strongly indicates differences based on viewing orientation. At Summit, HOIC are most commonly
observed in CAPABL's data set in precipitation and stratiform clouds. Results from CALIOP, e.g. Noel and Chepfer (2010), indicate HOIC are common in cirrus clouds. CAPABL does observe some HOIC in cirrus but rarely due to extinction caused by lower clouds. This suggests a viewing bias, both from the ground and from space, that impacts our understanding of ice crystal orientation. The unique diattenuation observations by CAPABL provide a ground-based capability to observe HOICs under different viewing conditions.

**6.2  Impact on Attribution of Cloud Effects on the Surface Radiation Budget**

In a similar method to data comparisons with LWP, comparisons of CAPABL data to observed downwelling and upwelling longwave (LW) and shortwave (SW) radiation fluxes have been performed. These comparisons elucidate the drawbacks of certain lidar methodologies and optimizes CAPABL's approach to provide a best estimate. The cumulative distribution functions of downwelling and upwelling LW and SW radiation measurements as well as the net radiation, defined as $Net = LW \downarrow$
$+ SW \downarrow - LW \uparrow - SW \uparrow$, are given in Figure 7 for CAPABL's merged best estimate data product, parsed into column types:





clear air, ice (with and without HOIC), and liquid bearing. Figure 7 shows some simple relationships that are examined for consistency with previous studies. The median value of downwelling LW radiation is higher for liquid clouds than it is for ice clouds, which is higher still than for clear air. This is expected based on many previous results including those of Curry et al. (1996); Shupe and Intrieri (2004); Miller et al. (2015). Likewise, the downwelling SW flux is highest for clear air and reduced for ice clouds, which is further reduced for liquid clouds. This shows the dominance of cloud visible optical depth by liquid clouds, which is well described by Shupe and Intrieri (2004); Stevens and Bony (2013); Miller et al. (2015). The upwelling LW measurements are highest for liquid cloud scenes, which can be understood based on the enhanced downwelling LW radiation and emission that scales with surface temperature to the forth power. Miller et al. (2017) showed warmer surface

temperatures occur with liquid clouds overhead. Finally, upwelling SW is simply the scaled version of the downwelling SW, scaled by surface albedo. These results are all expected and provide further validation that the CAPABL cloud identification procedure is acting as expected.

The median values of all distributions for the three CAPABL classification types, analog, photon counting, and merged, and all four radiation types and the net radiation are listed in Table 6. The merged column is our best estimate through signal

combinations so that a difference between merged and analog or merged and photon counting indicate limitations for those stand alone techniques. For example, the percent difference for the downwelling longwave radiation for clear air and ROIC is on the order of 5-10%. This difference for analog is attributed to difficulty measuring the whole column of ice especially in the polar summer with just orthogonal polarization retrievals. Due to its lower designed sensitivity, the analog clear air classification misses some ice clouds that contaminate the clear air classification. The difference for photon counting is attributed to

saturation. The ROIC classification is contaminated by low liquid clouds artificially raising the overall downwelling longwave effect. The same affects the downwelling shortwave measurements on the order of 10%. For photon counting measurements, contamination from liquid clouds lowers the downwelling shortwave component. For the total radiative components, the lack of sensitivity of the analog channel artificially raises the clear air radiative balance towards the values for ice clouds. For the photon counting component, saturation raises the ROIC radiative balance towards the values for liquid clouds. In all cases,

traditional lidar data used to attribute radiative fluxes will introduce large uncertainties based on the lidar's inability to measure the whole atmospheric column related to its limited dynamic range. Enhancing measurements as done in this work with analog and photon counting detection as well as non-orthogonal polarization retrievals allows for a more complete attribution of radiative effects linked to cloud properties.

A second method of analyzing the radiative importance of this work is to use literature values to estimate cloud radiative

effects. Figure 5 gives the FO of voxel types in the column above CAPABL. Using this FO, literature values such as those presented by Miller et al. (2015) can be used to estimate misattribution of cloud radiative effects. Figure 5 shows a difference of approximately 30% from analog to photon counting for liquid FO. This difference can be used to approximate an error in cloud radiative forcing using the results from Fig. 7 from Miller et al. (2015). Using an average difference of 30%, this time period of fractional occurrence of liquid clouds equates to an error in longwave cloud radiative effect of approximately

10 W/m$^2$. Miller et al. (2015) finds an average of 33 W/m$^2$ for cloud radiative forcing at Summit suggesting that using





conventional lidar approaches to infer radiative impacts could under-represent forcing by as much as one third. The CAPABL approach improves the situation significantly leading to better attribution of cloud effects on radiative fluxes.

## 6.3    Main Conclusions

This work has demonstrated three key points. The first point is that cloud phase classification by polarimetric lidar is sensitive not only to the cloud phase but to lidar design properties such as receiver polarization, detection schemes, and backscattered
signal count rate and, by extension, cloud macrophysical properties such as base height (or range) and optical depth. The second point is the utility of non-orthogonal polarization measurements to improve cloud classifications. By employing multiple planes of polarization in the lidar receiver, in the case of CAPABL four linear planes, the diversity in backscattered intensity may be handled more judiciously making the characterization of cloud types more accountable. This effectively spreads the required dynamic range of signals among the multiple polarization measurements. By measuring additional planes of polariza-
tion beyond what is required for geophysical retrievals allows the CAPABL system to self analyze limitations in a channel's performance, correct some of the behavior through non-orthogonal signal combinations, and optimize the use of the different channels for different cloud scattering conditions. In high dynamic range targets, like optically thick liquid-only or mixed-phase clouds, systematic errors can cause a misrepresentation in traditional polarization-sensitive lidars of liquid clouds as ice clouds. Here this is shown to occur on the order of 30% of the time for CAPABL but is correctable using the presented novel
polarization scheme. Finally, this work has analyzed the effects of lidar data in terms of radiative attribution. Using a particular detection system such as photon counting, orthogonal polarization measurements can dramatically mis-represent cloud radiative effect. Using radiation measurements from Summit, errors in attribution of radiative scenes related to cloud phase can be on the order of 10% of the net radiation. Using cloud fraction as an estimator with previously published radiative estimates of Miller et al. (2015) suggests an even higher 30% misattribution.

*Code and data availability.*    All data collected by the ICECAPS program is publicly available at: anonymous@ftp1.esrl.noaa.gov/psd3/arctic/summit/. Radiation data collected by the NOAA is publicly available at: ftp://aftp.cmdl.noaa.gov/data/radiation/baseline/sum/. The code developed to process the CAPABL data is available by request from the authors.

## Appendix A:  CAPABL's Nonlinear Photon Counting

CAPABL's photon counting system is subject to pulse pileup, as is the case with most photon counting systems. This pileup
results in detector pulses occurring too close in time for the counting system to uniquely identify individual pulses, resulting in systematic underrepresentation of photon count rate. The models introduced to correct this problem are based on the work of Donovan et al. (1993); Whiteman (2003); Liu et al. (2009) using a calibration data set taken during a clean air period at Summit in May 2015. The neutral-density filter was removed from the receiver optical path on a clear air day to increase the observed count rate and also extend the vertical range of calibration data. Data were concatenated based on the work of Newsom et al.





(2009) with the main difference being that profiles were background subtracted before analysis (note that this is the only case in this manuscript where such concatenation is performed). From these data, the analog profile is taken as the ideal count rate. These data are plotted in Fig. A1 with two correction methods fit to the data using a Levenberg-Marquardt nonlinear least

squares solver. These saturation models are given as

$$S_{obs} = \frac{S_0}{1 + \tau_{NP} S_0} \tag{A1}$$

and

$$S_{obs} = S_0 \exp\left(\tau_P S_0\right) \tag{A2}$$

referred to as non-paralyzable and paralyzable, respectively. The fit parameter for non-paralyzable is the deadtime $\tau_{NP}$ and for

paralyzable $\tau_P$.

To convert from the observed photon count number to observed photon count rate, the simple linear transformation

$$N_{obs} = S_{obs} \times S_{PP} \times T_{PB} \tag{A3}$$

is used where $N_{obs}$ is the observed photon count number per bin, $S_{obs}$ is the observed photon count rate per shot, $S_{PP}$ is the number of laser shots integrated per profile, and $T_{PB}$ is the two way travel time of light per range bin.

Inserting Eq. A3 into Eq. A1 and performing a propagation of error analysis, based on Taylor series expansion for standard error propagation assuming no data covariance, yields the shot noise error for the corrected photon count number per bin given as

$$\sigma_N = S_{PP} T_{PB} \sqrt{\frac{N_{obs}^4 \sigma_{\tau_{NP}}^2 + S_{PP}^2 T_{PB}^2 \sigma_{N_{obs}}^2}{\left(S_{PP} T_{PB} - \tau_{NP} N_{obs}\right)^4}}. \tag{A4}$$

Equation A4 indicates that the error in corrected photon count rate is a function of the count error $\sigma_{N_{obs}}$, which conform to

Poisson statistics, and the error in the model fit parameter $\tau_{NP}$. This error is estimated during the fitting procedure using the fit confidence bounds. Note that if and only if $\tau_{NP}$ is exactly zero (i.e. $\tau_{NP} = 0$ and $\sigma_{\tau_{NP}} = 0$) will the counting error be simply $\sigma_{N_{obs}}$.

The calibration data used for this analysis is presented in Fig. A1. As each measurement is subject to some measurement error, Poisson counting error for photon counting and electrical noise for the analog detection, this fit was calculated using the

SNR as a data weight such that higher SNR data are given higher weights. The results of this weighted analysis indicate that the dead time is approximately 0.1 ns higher than the unweighted analysis which ignores measurement errors in the fit.

*Author contributions.* R. Stillwell (RS) developed the processing code for CAPABL and MPL data, the data merging procedure, and performed the multi-sensor validation and radiation analysis. M. Shupe (MS) provided MMCR data. D. Turner (DT) provided the MWR and MPL raw data and performed the MWR optimal estimation retrievals. J. Thayer (JT) and Ryan Neely (RN) served as advisors for RS for





CAPABL specific technical tasks and RN, MS, and DT contributed scientific context. The ICECAPS instrument is maintained by technicians
from Polar Field services in close coordination with RS, RN, MS, and DT. RS prepared the manuscript with contributions from RN, JT, MS,
and DT.

*Competing interests.*    The authors declare no competing interests.

*Acknowledgements.*    This material is based upon work supported by the National Science Foundation Graduate Research Fellowship Program
under grant DGE 1144083 and National Science Foundation grants AON 1303864, PLR-1303864, PLR-1303879, PLR-1314156, and ATM-
0454999. Ryan Neely is funded by the National Centre for Atmospheric Science. The authors would like to thank the staff and science
technicians at Summit, especially Hannah James and Samuel Dorsi, as well as Polar Field Services for their support and dedication to help
collect data and maintain instrumentation. The authors would also like to thank David Longenecker for providing radiation data. ICECAPS
MPL data was provided by the US Department of Energy Atmospheric Radiation Measurement Program while MMCR data was provided
by the NOAA Earth System Research Laboratory.





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





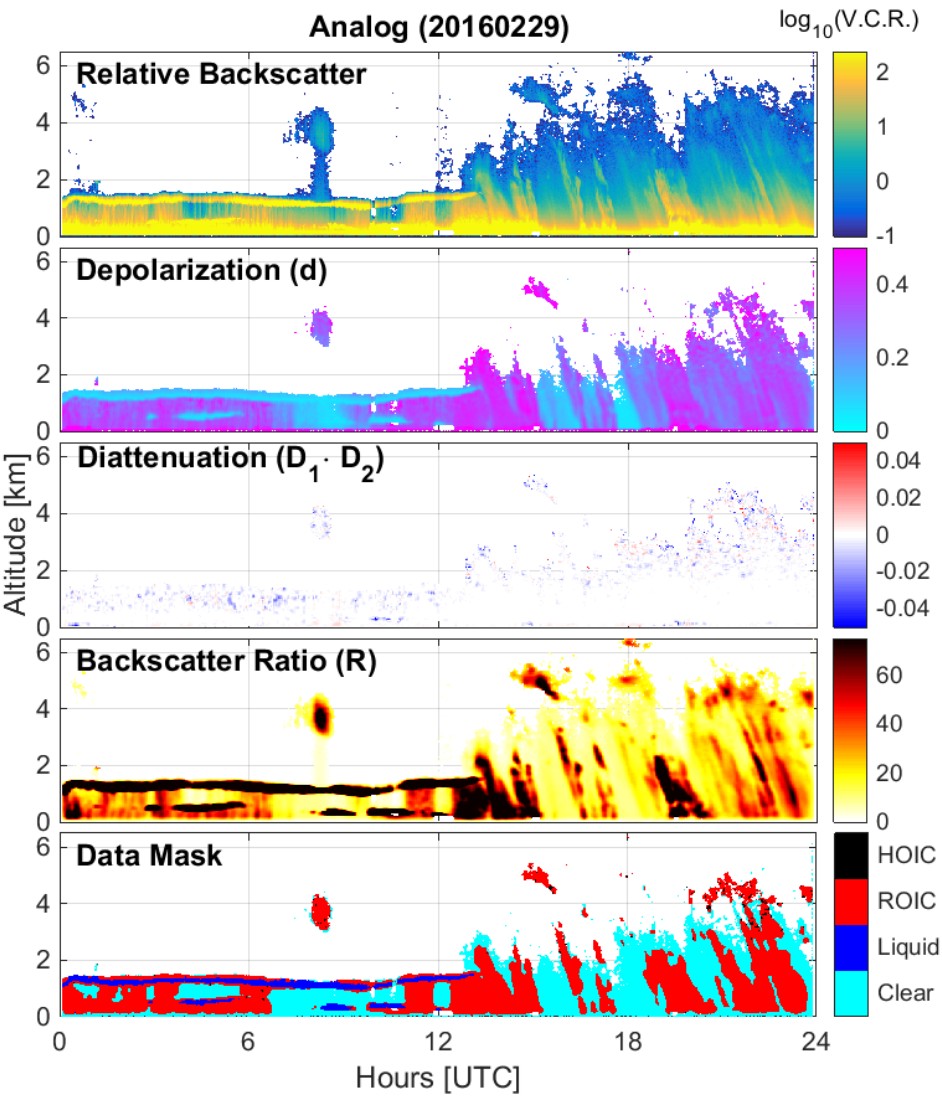

**Figure 1.** Analog data from the CAPABL system for February 29, 2016. Relative Backscatter is the summation of background subtracted parallel and perpendicular voltages converted to a virtual count rate (V.C.R.) in MHz. The total backscatter color bar is given from 100 kHz to 250 MHz on a logarithmic scale. Depolarization is calculated as given in Eq. 7. Diattenuation is calculated as given in Eq. 8 and multiplied to $D_1 D_2$. Backscatter ratio is calculated by performing a Klett inversion and using ICECAPS radiosonde data (launched at 2400 UTC and 1200 UTC daily) to calculate a molecular extinction component (Klett, 1981). The data mask given is calculated using rules described in Sect. 3.





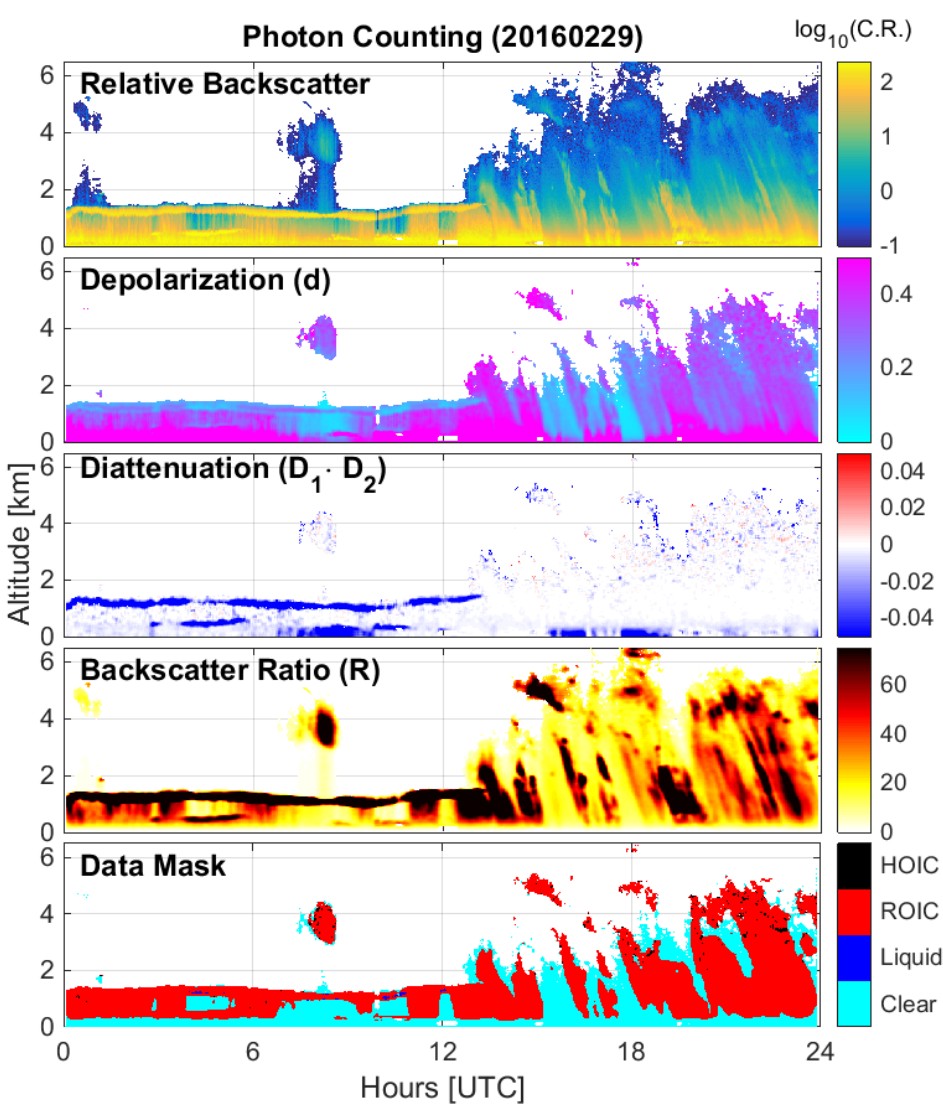

**Figure 2.** Same as Fig. 1 except photon counting data are shown.



**Figure 3.** CAPABL data from July 2015 to October 2015 binned into liquid, ice, or clear air. The median is indicated by a line through the box, the 25th to 75th percentile ranges complete the box and the whiskers extend to the 5th and 95th percentiles. PC, SCPC, and N.O. stand for photon counting, saturation photon counting, and non-orthogonal, respectively. The channel sensitivity can be seen looking at the clear voxels where analog is expected to be less sensitive than PC and orthogonal less sensitive than non-orthogonal.





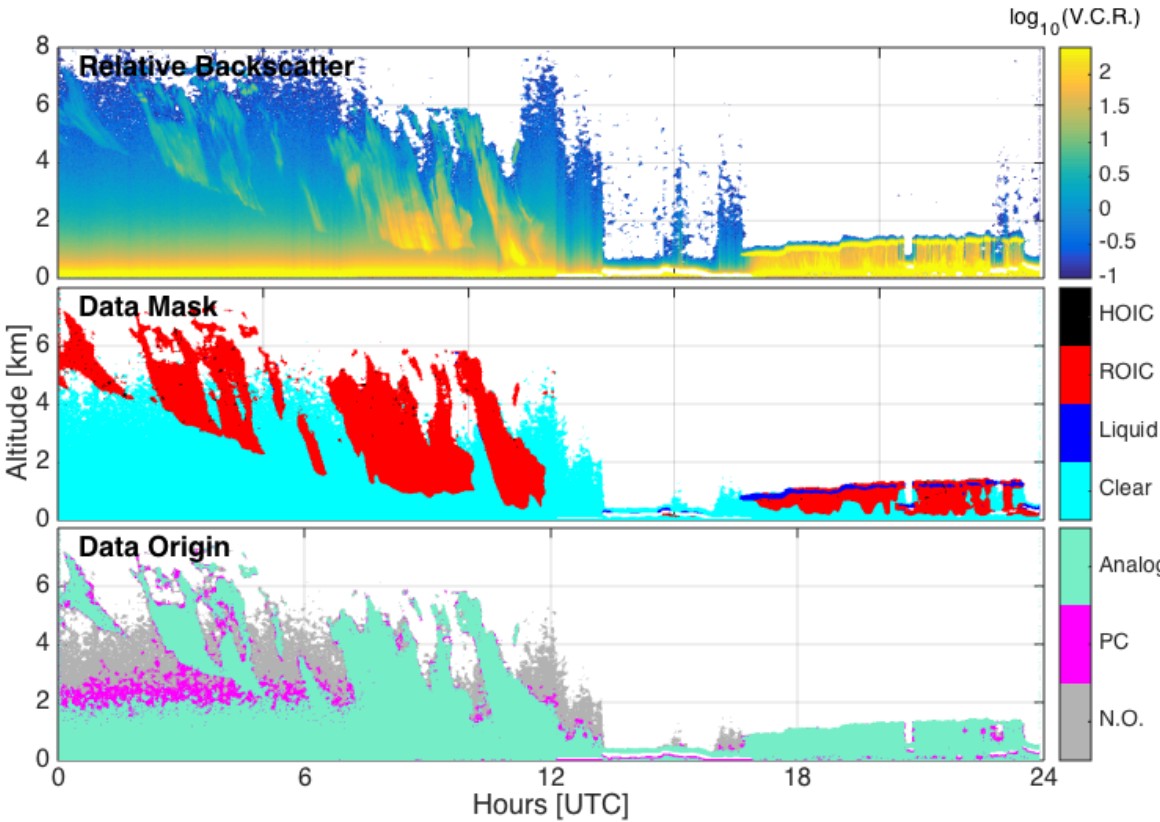

**Figure 4.** A sample of the CAPABL merged data product from August 22, 2016. The top panel shows total analog backscatter for the whole day in log base 10 signal intensity. The middle panel shows the merged data product. The bottom panel shows the origin of each voxel. Analog indicates orthogonal processing with analog data, PC indicates orthogonal processing with photon counting data. All non-orthogonal types are lumped together as N.O.





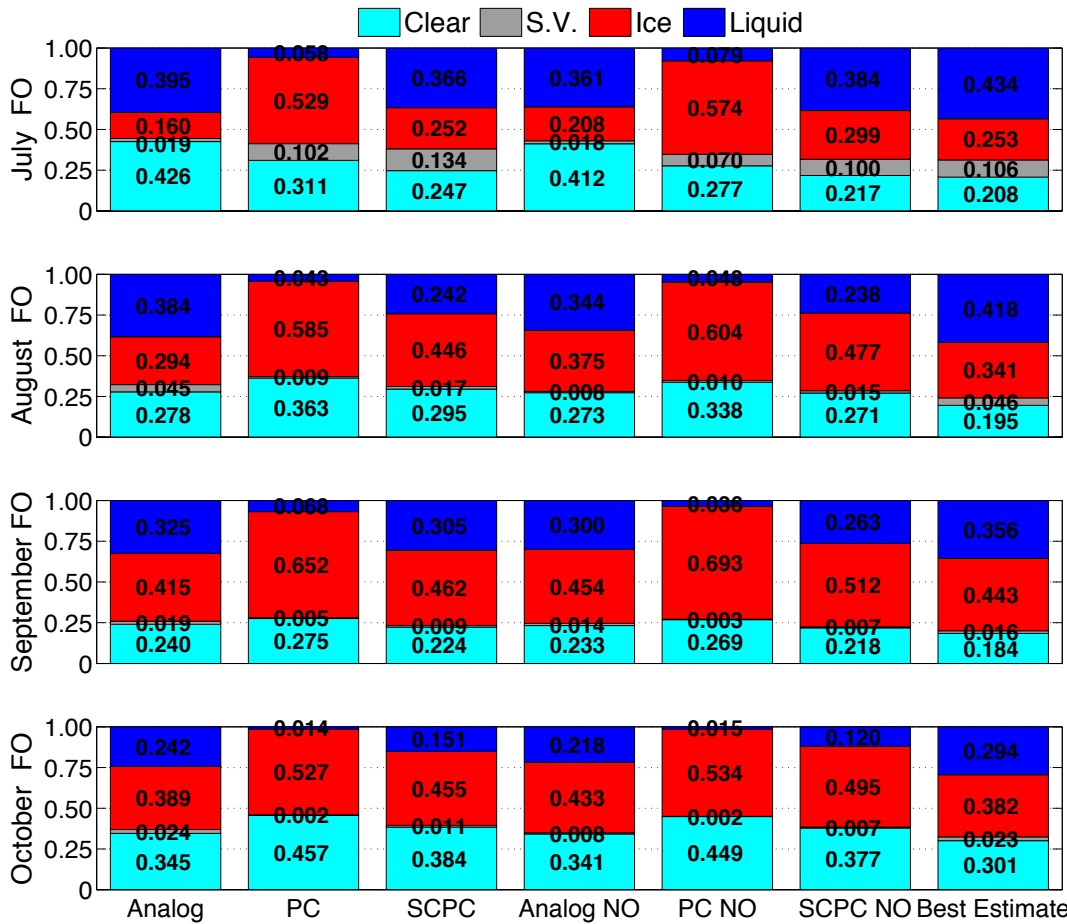

**Figure 5.** Fractional occurrence (FO) of each pixel type in the column for July 2015 to October 2015. To be labeled clear, the column must lack all sub-visible, ice, and water pixels. To be labeled sub-visible, the column must lack ice or water pixels. To be labeled as ice, a column must lack water pixels. If a column contains a water pixel, the column is labeled as liquid. The FO is given for each bar rounded to the nearest thousandth.



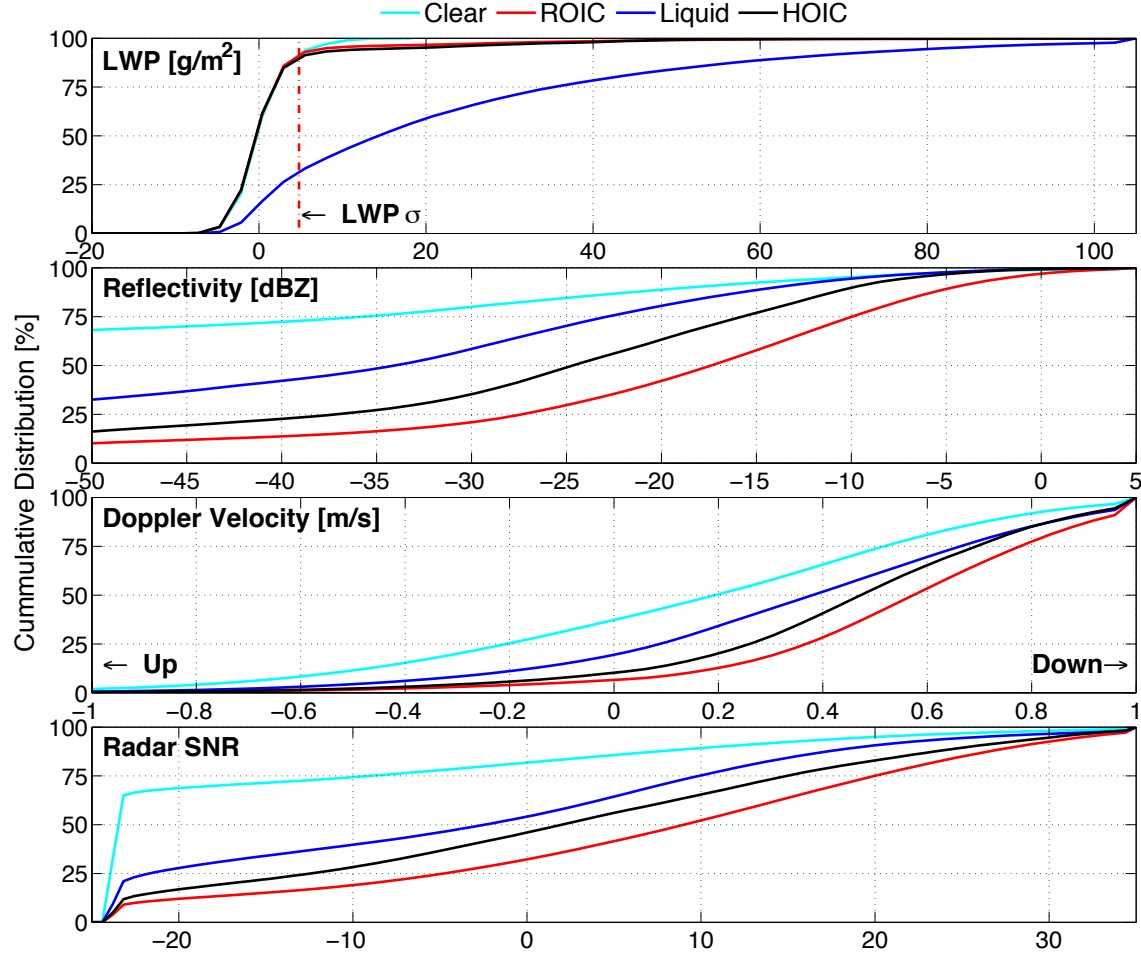

**Figure 6.** Cumulative distribution functions of co-located ICECAPS data parsed by CAPABL classification type. All data from July 2016 to December 2016, approximately 54 million radar voxels for each Doppler moment and 148,000 MWR column measurements, are collected and identified. Note that the average LWP uncertainty is given for the entire study period and that here a positive mean Doppler velocity is defined towards the zenith pointing radar system or downwards. For the LWP uncertainty, assuming an effective radius of $r_e = 10$ µm, a density of water of $\rho = 1000 \ \mathrm{kg/m^3}$, and using the approximate relation $LWP = 2\tau r_e \rho/3$ yields a threshold for optical depth of $\tau = 0.75$.



**Figure 7.** Cumulative distribution function of downwelling and upwelling radiation data at the surface parsed by CAPABL column classification type.





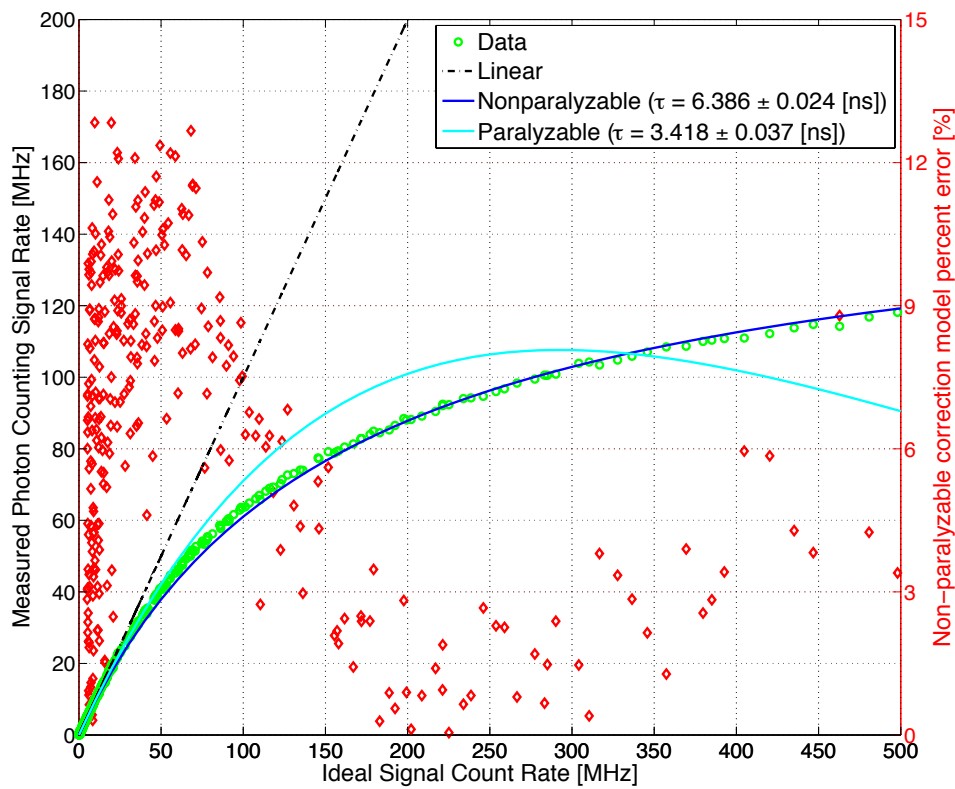

**Figure A1.** Saturation analysis of the CAPABL photon counting channel using the theory developed by Donovan et al. (1993); Whiteman (2003); Liu et al. (2009). The ideal signal count rate is found by normalizing the analog detection channel to the photon counting channel in a region where both are acting linearly which is about 1 MHz count rate. The measured count rate is then taken directly from photon counting measurements. The paralyzable and non-paralyzable models are then fit using a Levenberg-Marquardt weighted non-linear least squares fitting algorithm of the observed calibration data. The $1\sigma$ confidence bound is given for each dead time fit parameter. Finally, the percent error of the correction model is given relative to the ideal count rate on the right ordinate as diamonds.



**Table 1.** CAPABL current system specifications. Polarization purity and polarization rejection are measured quantities. Polarization purity is measured with a 100,000:1 Glan-Taylor polarizer.

| Transmitter | Receiver | Signal Processing |
| --- | --- | --- |
| Big Sky Laser Ultra flashlamp pumped Nd:YAG | Schmidt Cassegrain Telescope | Combined analog and photon counting acquisition |
| Wavelength: 532.3 nm | Receiver Aperture: 20.8 cm | Data system: |
| Pulse Energy: 60 mJ | Filter Bandwidth: 0.3 nm | Licel Transient Recorder TR20-12 Bit |
| Pulse Rate: 15 Hz | Channels: 1 | Range bin size: 7.5 m |
| Twin Head | Field of View: 1.4 mrad | Integration time: 5 sec |
| Polarization Purity: $> 123 : 1$ | Polarization Rejection: $> 800 : 1$ | PMT: Hamamatsu R7400U-03 |
| | Linear Polarizations Observed: 4 | |





**Table 2.** A summary of the data processing steps taken to create the data masks desired for CAPABL. The processing for each data type: Analog (An), Photon Counting (PC), and Saturation Corrected Photon Counting (SCPC), is constant except where noted. Note that the depolarization and diattenuation error equation are calculated per standard propagation of error techniques taking a Taylor series expansion of Eq. 7 and Eq. 8.

|   | Processing Step | Details |
|---|---|---|
| 1) | Time integration | To a constant 20 sec resolution |
| 2) | Spatial integration | To a constant 30 m resolution |
| 3) | Saturation correction (PC Data) | Creates SCPC level |
| 4) | Background subtraction | |
| 5) | SNR filter | |
| 6) | Speckle filter | $5 \times 5$ surrounding box |
|   |   | $> 75\%$ data already removed $=$ bad |
|   |   | $> 25\%$ data available $=$ good |
| 7) | Calculate polarization properties | Depolarization and depolarization ratio per Eq. 7 and 9 |
|   |   | Depolarization and depolarization ratio error per error propagation of Eq. 7 and 9 |
|   |   | Diattenuation per Eq. 8 |
|   |   | Diattenuation error per error propagation of Eq. 8 |
|   |   | Backscatter ratio ($R$) per (Klett, 1981; Neely et al., 2013) |
| 8) | Remove non-physical values | Values outside $0 \leq \delta_O \leq 1$ |
|   |   | Values outside $0 \leq \sigma_{\delta_O} \leq 0.4$ |
|   |   | Values outside $-1 \leq D \leq 1$ |
|   |   | Values outside $0 \leq \sigma_D \leq 0.2$ |
| 9) | Calculate base mask | Clear: $1 \leq R < 2.6$ |
|   |   | Aerosol: $2.6 \leq R < 6.5$ |
|   |   | Cloud: $R \geq 6.5$ |
| 10) | Calculate phase mask | Liquid: cloud voxels with $0 \leq \delta_O \leq 0.11$ |
|   |   | Ice: cloud voxels with $\delta_O > 0.11$ |
| 11) | Calculate orientation mask | Random: ice with $0 \leq D_1 D_2 \leq 0.01$ |
|   |   | Preferential: ice with $D_1 D_2 \geq 0.01$ and $\sigma_D \leq 0.05$ |
|   |   | Saturation: ice with $D_1 D_2 \leq -0.01$ |





**Table 3.** Hardware comparison of relevant CAPABL and MPL lidar specifications. The resolutions quoted are limited in range by the MPL afterpulse calibration data and in time by the CAPABL scan rate. The resolutions presented are as close as the data can be processed before linear interpolation of MPL data to CAPABL's data grid. Effective power aperture product is reduced for CAPABL by the receiver attenuation by a factor of 1000.

| Specification | CAPABL | MPL |
|---|---|---|
| Laser Power [W] | 0.3 | 0.02 |
| Receiver Attenuation [OD] | 3 | 0 |
| Telescope Diameter [mm] | 208 | 178 |
| Effective Power/Aperture Product [$W \cdot mm^2$] | 10.2 | 497 |
| Polarizations | 4 | 2 |
| Range Resolution [m] | 25.98 | 30 |
| Polarization Scan Resolution [s] | ≈82 | 80 |





**Table 4.** Radar operational mode configuration settings. The radar cycles between 4 modes of which only the cirrus and general modes are used in this work. The modes are cycled such that the general mode is every 4th measurement and the cirrus mode is every 8th at a cadence of approximately 0.5 sec per mode.

| Radar Mode | General | Cirrus |
|---|---|---|
| Average power [W] | 0.5353 | 7.146 |
| Intra-pulse period [ms] | 96 | 115 |
| Pulse width [ns] | 583 | 583 |
| Number of coded bits | 0 | 16 |
| Number of coherent averages | 5 | 6 |
| Range resolution [m] | 87.5 | 87.5 |





**Table 5.** Confusion matrix of CAPABL and MPL processed data. The diagonal shows agreement, highlighted by bold text. The last row and last column indicates one instrument had data removed by quality control steps, also highlighted in italics. Cells B and C indicate enhanced sensitivity by CAPABL processing and cells E and I indicate enhanced sensitivity by the MPL processing. Cell P indicates both instruments lack data implying that much of the data missed is in a regime not reachable via lidar (i.e. large optical depth). Three sets of data are given in each cell, which are identified by the last column. The first line of each cell covers the time period July 1st - July 31st, 2016. The second line of each cell covers December 1st - December 31st, 2016. The final row of each cell covers July 1st - December 31st, 2016.

|  | CAPABL Clear | CAPABL Liquid | CAPABL Ice | CAPABL Filtered | Time Period |
|---|---|---|---|---|---|
| MPL Clear | A) **69.7%** | B) 37.0% | C) 62.2% | D) *3.4%* | July |
|  | **97.7%** | 64.9% | 78.9% | *74.5%* | December |
|  | **83.2%** | 41.8% | 63.9% | *35.1%* | All |
| MPL Liquid | E)0.3% | F) **56.3%** | G) 5.5% | H) *0.1%* | July |
|  | 0.0% | **26.3%** | 0.2% | *0.0%* | December |
|  | 0.4% | **47.9%** | 2.0% | *0.2%* | All |
| MPL Ice | I) 0.2% | J) 3.7% | K) **29.4%** | L) *0.5%* | July |
|  | 0.2% | 8.2% | **20.2%** | *0.3%* | December |
|  | 1.4% | 8.9% | **31.7%** | *1.1%* | All |
| MPL Filtered | M) *29.9%* | N) *3.0%* | O) *3.0%* | P) **96.0%** | July |
|  | *2.1%* | *0.5%* | *0.6%* | **25.2%** | December |
|  | *15.1%* | *2.5%* | *2.4%* | **63.7%** | All |





**Table 6.** Median values of the probability distribution function for each data processing type for each radiation component. Each radiation component is measured at the surface in units of $W/m^2$. Total flux is calculated using the relation: $Total = LW\downarrow + SW\downarrow - LW\uparrow - SW\uparrow$.

| | Type | Merged | Analog | PC | % Difference: Merged and Analog | % Difference Merged and PC |
|---|---|---|---|---|---|---|
| **Downwelling Longwave** | | | | | | |
| | Clear | 153.3 | 162.1 | 155.3 | 5.6 | 1.2 |
| | ROIC | 175.1 | 181.2 | 194.3 | 3.4 | 10.4 |
| | Liquid | 222.9 | 224.1 | 213.5 | 0.5 | 4.3 |
| | HOIC | 179.8 | 179.2 | 179.9 | 0.4 | 0.0 |
| **Upwelling Longwave** | | | | | | |
| | Clear | 190.6 | 209.8 | 204.3 | 9.6 | 6.9 |
| | ROIC | 203.2 | 206.0 | 214.3 | 1.4 | 5.3 |
| | Liquid | 232.0 | 232.9 | 227.6 | 0.4 | 2.0 |
| | HOIC | 207.8 | 206.4 | 207.5 | 0.6 | 0.1 |
| **Downwelling Shortwave** | | | | | | |
| | Clear | 524.1 | 571.7 | 539.8 | 8.7 | 3.0 |
| | ROIC | 381.8 | 356.6 | 342.8 | 6.8 | 10.8 |
| | Liquid | 342.5 | 342.4 | 379.2 | 0.0 | 10.2 |
| | HOIC | 426.2 | 418.9 | 418.0 | 1.7 | 1.9 |
| **Upwelling Shortwave** | | | | | | |
| | Clear | 422.9 | 462.6 | 426.6 | 9.0 | 0.9 |
| | ROIC | 321.2 | 309.4 | 294.6 | 3.7 | 8.6 |
| | Liquid | 293.3 | 293.0 | 323.8 | 0.1 | 9.9 |
| | HOIC | 359.2 | 352.4 | 353.2 | 1.9 | 1.7 |
| **Total** | | | | | | |
| | Clear | -24.3 | -23.7 | -25.0 | -2.4 | -2.7 |
| | ROIC | -15.5 | -14.7 | -9.0 | -4.9 | -53.3 |
| | Liquid | -1.1 | -0.7 | -4.0 | -48.2 | -112.2 |
| | HOIC | -17.4 | -17.2 | -16.1 | -1.1 | -7.4 |