# Peer review of "Improved Cloud Phase Determination of Low Level Liquid and Mixed Phase Clouds by Enhanced Polarimetric Lidar"

_Atmospheric Measurement Techniques, 2017_

## Referee Comment (RC1) · Anonymous Referee #1 · 31 Oct 2017

This paper describes a novel processing technique, using data from the CAPABL instrument at Summit, Greenland, which incorporates both the instruments' multiple polarization capabilities, as well as its photon counting and analog detection modes. Radiation, radar, and liquid water path measurements taken at the same site are used to validate this method.

This paper is well written; it provides an excellent background and theory of the measurements, as well as good description of the different instruments (Table 2 is very useful as well). It is clear that the authors are deeply familiar with this instrument and the portrayed methodology. The latter is new (to the extent of my knowledge), interest-

ing, and significant (to relevant lidar instruments).

However, there are several important issues that should be addressed. Therefore, I recommend this work for publication after the following issues will be treated:

Major comments:

-M1: Section 3.2: I understand that this is not a HSRL or a Raman lidar, where the lidar ratio can be extracted from the measurements. However, In Nott and Duck (2011) the variability of the lidar ratio is quite large, and is more towards 20 on average (see also Thorsen and Fu, 2015, DOI: 10.1175/JTECH-D-14-00178.1). There are several studies where lidar ratio of 10 was concluded (for certain aerosols), but these should be cited directly, and not through Nott and Duck (2011). This is a very delicate point, as the variability of the lidar ratio of different hydrometeors can significantly affect the backscatter ratio extraction, which is needed for the classification procedure. As the authors rely on this assumption for quantitative analysis, I wonder to what extent the results would change by using other lidar ratio values. Was there any other reason such a low lidar ratio was used (e.g., problematic resolved parameters, etc.)?

-M2: Section 3.3: Lidars operating at the same wavelength and probe the same air volumes should produce in theory the same output parameters. However, differences in the configuration and electronics, as well as in the data processing (e.g., in DABUL vs. CAPABL), all induce deviation in the resolved parameters, in particular in LDR values. In addition, fixed thresholds using only the LDR and backscatter values can be quite problematic in general for classification, as they don't take into consideration the atmospheric variability (I suggest looking at Figure 3 in Thorsen and Fu, 2015, and Figure 4 in Luke et al, 2010, doi:10.1029/2009JD012884), and the possible ambiguity of lidar returns (which is controlled to a certain degree using multiple polarizations). These effects are emphasized when utilizing a constant lidar ratio (which results in backscatter coefficient uncertainty, and hence, backscatter ratio uncertainty). Change of the LDR threshold to high values won't alter by much the FO as most lidar returns

with high backscatter values are located at low LDR (as seen in the figures mentioned above), so the description in p. 11, l. 23-29 is not surprising. The region where the sharp shifts in FO of liquid and ice at low LDR are relevant, and I think an analysis of this region, and how much changing the thresholds would affect the results, should be presented here (or in a supplementary material / Appendix).

-M3 Sections 3.4, 4 and 5: examination of Figures 1, 2 and 4 increase my concerns in M1 and M2, as the resolved (upper, at ∼1.5 km) liquid layers are much thinner than I would expect, when inspecting the LDR and relative backscatter (the signal becomes extinct due to the optically-thick liquid, which extends deeper than deduced). I suspect that this is a result of the constant lidar ratio, and consequently, the backscatter ratio threshold for liquid detection. Thus, based on this plot, the results seem to underestimate the liquid amount. This underestimation is not detected and remains "under the radar" in the validation section (e.g., figures 6 and 7), as columns are treated there (and not voxels), i.e., it is enough that a single ice/liquid voxel is kept after the filtering for the column to be treated as ice/liquid bearing. It might be relevant to these figures in cases of liquid containing air-volumes above intense precipitation, where the signal is strongly (but not completely) attenuated, but these cases are relatively rare at Summit. In addition, the weak overlap between the analog and PC percentiles shown in Figure 3 makes me suspect that there is a "blind" zone around 1.5 km, where liquid voxels might saturate the PC while the signal having below threshold SNR (due to extinction/tenuous layer) in the analog.

-M4 Section 5: In continuation of the latter, missed detection analysis is needed in this paper as well, to confirm that hydrometeors of a certain type were not missed by CAPABL (as long as the signal is not extinct), thus strengthening the reliability of the study's methodology. E.g., CDF of periods when the MWR detected liquid above the uncertainty level but the CAPABL did not detect any liquid voxel (directly related to significant LWP levels missed in ∼10% of ROIC, HOIC, and clear, as stated in p.19 l.24-32), percentages of missed MMCR bins above SNR of -14 dB (i.e., hydrometeors,

most likely ice, after Shupe, 2013) which were missed by CAPABL.

-M5 Section 5: What I get from Table 5 is that the MPL is not a good instrument to validate the CAPABL retrievals, but to show CAPABL's superiority. The low percentages in boxes G and J, and the high ones in B and C, mean (as mentioned in the text) that it is not possible to genuinely compare the two instruments. This leaves the validation merely to the MWR and MMCR. I suppose that the delicate fixed liquid/ice determination thresholds have (a more significant) role in the MPL data analysis as well.

Minor comments:

-m1 p.9 l.25: Add 'volume pixel' in parentheses after 'voxel'.

-m2 p.14 l.9: Change 'is' to 'are'

-m3 p.15 l.0 and the entire paper: Please be consistent and use either 'Fig.' or 'Figure'.

-m4 p.16 l.27-28: Please provide a citation for this low LWP uncertainty.

-m5 p.19 l.17: Change 'has' to 'have'

-m6 p.20 l.2-3: Please provide a citation for this argument regarding the cirrus mode artifacts.

-m7 p.20 l.4-7: Could there be a height effect of CAPABL measurements as well (e.g., due to different operated modes, varying lidar returns' true lidar ratio below a certain altitude, which affect the integrated column above, etc.)? The two possibilities stated by the authors might be valid, but it will be suitable to mention them after comparison of both MMCR modes above 3 km will be performed.

-m8 Figure 1: Please consider changing the colorbar around 0 to grey, as it is impossible to distinguish between missing or "bad" lidar returns and values near zero. In addition, consider extending the scale of the relative backscatter panel, as it is impossible to separate the intense lower liquid layers' returns from the noisy background.
I suggest adding (twice) daily sounding temperature profile to the plot, to enable the examination of the reliability of the HOIC classification given the temperature range.

-m9 Figure 4: Similarly, consider extending the scale of the relative backscatter panel, as it is impossible to separate the intense lower liquid layers' returns from the noisy background.

-m10 Figure 6: Please have a citation for the approximation given in the caption.

---

## Referee Comment (RC2) · Anonymous Referee #2 · 7 Nov 2017

**1 General remarks**

This is a review of the manuscript "Improved Cloud Phase Determination of Low Level Liquid and Mixed Phase Clouds by Enhanced Polarimetric Lidar" submitted by Robert A. Stillwell et al. to Atmospheric Measurement Techniques Discussions. This paper appears to be a revised resubmission of the original version with the title "Low-Level, Liquid-Only and Mixed-Phase Cloud Identification by Polarimetric Lidar", which was submitted by Robert A. Stillwell et al. to AMTD in 2016.

This manuscript delves further into the application of depolarization lidar meaasurements to retrieve cloud phase and cloud particle orientation in the Arctic. While the phase determination of Arctic low-level clouds are a challenging task, current phase statistics of these clouds still lack the accuracy to constraint the Arctic energy budget. Here, polarization-sensitive lidars can help to discriminate ice from liquid clouds since ice particles are depolarizing, while liquid particles are not.

To this end, the manuscript makes use of the depolarization measurement capability of the CAPABL lidar at Summit, Greenland. Specifically, uncertainties in the determination of thermodynamic phase of clouds in the Arctic caused by depolarization measurement errors are discussed.

The key innovations of this manuscript are focused on two methods to deal with the problem of detector saturation for polarization measurements using the CAPABL lidar. First, the generalization to non-orthogonal polarization channels within the approach described by Neely et al., 2013, who added a fourth polarization channel to detect saturation effects. Second, the comparison and the combination of photon counting and analog signals to enhance the dynamic range for cloud phase determination. Statistics conducted over several months show, that up to 30% of optically thick liquid or mixed-phase clouds can be misclassified when using only photon counting or analog signals.

The manuscript starts off rather technically, before it introduces the two new methods for cloud phase determination. The differences in cloud phase when using only photon counting or analog detection should definitely get the attention of a broader scientific audience.

In this regard, this work provides an important contribution to current and future research and is worth to be published as it helps to address the current bias in surface flux estimates in the Arctic caused by the undetected liquid phase of low-level boundary clouds. In my opinion, however, the manuscript lacks of five major issues which have to be addressed in detail before publishing the manuscript. Below, I compiled a list of comments which have to be considered in a revised version of the paper.

**2 Major comments**

**1. Structure of the manuscript**

The manuscript is some times hard to read, since some sections are not well embedded into the overall structure. I found mainly two reasons for this:

– First, section 2.1) "Polarization Measurements and Mueller Formalism" is very long and detailed while your objective (phase determination of Arctic low-level clouds) and your approach (combination of analog and digital detectors for cloud polarimetry) become stifled by too much formalism. This makes it hard for the reader to understand and appreciate the novelty of your approach. I agree, that polarimetric measurements and your non-orthogonal retrieval can not completely be discussed without an introduction to the Mueller Formalism. But considering the length of this section I would prefer an introduction to the problem caused by sensor saturation and into your approach to derive a cloud phase fractional occurrence by combining analog and digital detector measurements. While the introduction of a fourth polarization channel with opposite sensitivity to detector saturation was already explained in Neely et al., 2013, your work should focus more on your novel merged best estimate cloud product.

There are various passages throughout the text, which should be moved (or even get their own subsection) in front of section 2.1) to motivate its lengthy formalism:

  – Page 7, Line 13-21

*Observed depolarization ratios are a function of atmospheric scattering, optical system setup, and recording systems. Traditional two-channel polarization systems can not unambiguously measure atmospheric depolarization without additional information. Separating atmospheric depolarization from systematic effects is non-trivial. Alvarez et al. (2006) show, for example, how to calibrate differential detector sensitivity and receiver cross-talk, while Hayman and Thayer (2009) show how to remove depolarization effects caused by receiver optical retardance and scattering. However, recording systems that are subject to saturation, or underrepresentation of signal strength compared to incident irradiance, can also cause depolarization ratio effects, which are not constant in range and can not be calibrated using methods like that presented in Alvarez et al. (2006) or Hayman and Thayer (2009). Depolarization effects related to saturation couple polarization measurements with terms in the SVLE like cloud base height, range, and optical thickness through signal intensity measurements.*

– Page 10, Line 18-23

*By design, CAPABL uses 4 polarization channels to measure 3 elements of the scattering Mueller matrix: $F_{11}$, $F_{12}$, and $F_{33}$ with one additional measurement to monitor saturation. If saturation is not an issue, any 3 of the 4 channels may be used for the inversion of polarization properties. Thus, the utility of the generalization given in Sect. 2 is that the 3 signals with the least error can be used at any time. For example, the 3 strongest signals for measurements of high ice clouds where backscattered signals are weaker or the 3 weakest measurements for low liquid clouds where the backscattered signal is stronger. Using non-orthogonal polarizations allows the dynamic range between polarization components to be accommodated and optimized.*

– Page 11, Line 8-15 Covers data processing and not classification and therefore belongs into section 3.2) (somewhere around P9, Line 27)

- **–** Page 15, Line 1,4 ("First, ..." [...] "Second, ..." ) While this paragraph is a nice and comprehensible discussion of Figure 4, the two problems of photon counting and analog detection should be mentioned much earlier!
- **–** Page 22, Line 21-23: This is a very important summary sentence which I would like to see already in the abstract!

– Second, I suggest to drastically reduce or move several equations and paragraphs from section 2.1) into the appendix. This was already suggested by a referee (RC2) for the original submitted manuscript (amt-2016-303), but not implemented by the authors. In my opinion, this section should briefly describe the "Generalization of Neely et al., 2013" to non-orthogonal polarization channels and should not discuss the Stokes vector lidar equation or its components. For this reason, I can only reiterate the comment already made by RC2 that the *few equations and definitions that could be considered relevant to the data analysis (e.g., Eqns. 7, 8, 9 and 10)* can remain, while Eqns. 1-6 and the corresponding text should be removed or moved to the appendix.

Moreover, I would recommend to change some section headlines to improve the clarity of the manuscript:

- **–** Page 6, Line 13: New subsubsection 2.2.1) "Depolarization"
- **–** Page 7, Line 12: Subsection to subsubsection 2.2.2) "Diattenuation"

**2. **Lidar ratio**

Section 3.2, Page 10, Line 16:

*The inversion technique of Klett (1981) is used to calculate total scattering coefficient as described by Neely et al. (2013). A lidar ratio of 10 is assumed, following the review of Nott and Duck (2011) and references therein, to convert the total extinction derived by the Klett inversion to total backscattering coefficient.*

The constant lidar ratio of 10 used in the Klett inversion technique seems to be quite small when used for complete profiles. Theoretical and measured values for most ice crystals are much higher with average lidar ratios of around 22-28 (see e.g. Larroza et al., 2013, doi:10.5194/amt-6-3197-2013). Even liquid cloud droplets have an average lidar ratio of around 18 and rarely go below 10 (see e.g. Thorsen and Fu, 2015, doi:10.1175/JTECH-D-14-00178.1). This low lidar ratio should cause a low bias in estimated extinctions and therefore should lead to a low bias in backscatter ratios. This should, in particular, have an influence on the distribution of clear sky, sub-visible and aerosol voxels with backscatter ratio thresholds of 2.6 and 6.5. Could you elaborate your decision to use a lidar ratio of 10 and/or check if your results change for other lidar ratio values?

**3. Multiple scattering**

In this resubmitted version of the manuscript, the issue of multiple scattering on depolarization measurements is not mentioned at all. While the authors made it clear in a previous answer that "the effect is consistent across all 3 data sets" and thus "is recognized for future work but not implemented in the masking scheme", the effects of multiple scattering cannot be left unmentioned! Multiple scattering is known to enhance depolarization ratios in liquid clouds, which can lead to a misclassification as ice clouds. With the different field of views of 100 $\mu$rad for MPL and 1.6 mrad for CAPABL I would expect to see differences in measured polarizatiion ratios in non-saturated signal regions. Can you quantify this difference and make an assessment if this affects your classification?

**4. Choice of diagrams**

The author's choice of diagrams is sometimes not very helpful in guiding the reader to comprehends and verify the statements made. The following two figures should be reexamined:

– **Figure 3:**

Statistics for liquid, ice and clear air voxels using the different detection techniques are illustrated in terms of the median and percentile detection height. While this plot shows significant differences in the median height of liquid voxels, no clear trend is visible for ice or clear voxels. I do not agree with the authors' statement that "As a result of the increased sensitivity, the median altitude of the clear-sky data shifts upwards". Detection sensitivity (is there a cloud?) and efficiency (how much of it was detected?) can and should be tested in comparison with different instruments (e.g. MPL and MMCR). Using the median detection height is only useful to explore the impact on cloud height statistics, not detection sensitivity. For this reason I would remove the sub-figures for ice and clear voxels and only focus on the misclassification of liquid clouds. In contrast, Figure 5 does a very good job to detect differences in fractional occurrence of liquid or ice clouds, which could be easily compared to different datasets (e.g. Mioche et al., 2015, doi:10.5194/acp-15-2445-2015).

**– Figure 7:**

Page 19, Line 19-21:

*Note that though they contain and represent the same data, this work will choose to represent instrument comparisons in terms of their cumulative distribution functions as opposed to the probability density function. Both facilitate comparisons of large quantities of data but cumulative distribution functions allow simple comparisons of differences of shape and median whereas the probability density function allows for investigations of modes and biases.*

While this might me a minor issue, in my opinion, CDFs are harder to evaluate. While making differences more visible by using CDFs in Figure 7 might be acceptable, their significance is not discussed at all. In the end, Table 6 is much more valuable for comparisons with other studies. It would be nice to see PDF versions of Figure 6 and Figure 7 in your answer to decide if CDFs are really helpful here.

**5. Discussion of radiative impact**

My last objection is the way how the "Impact on Attribution of Cloud Effects on the Surface Radiation Budget" in Section 6.2) is discussed. While the observation that

Page 22, Line 32-33

*The median value of downwelling LW radiation is higher for liquid clouds than it is for ice clouds, which is higher still than for clear air*

or

Page 22, Line 1

*Likewise, the downwelling SW flux is highest for clear air and reduced for ice clouds, which is further reduced for liquid clouds.*

can be considered a scientific fact, a comparison of absolute values from the cited literature would be insightful. Moreover, it would be nice to use the measured absolute flux values from Table 6 when comparing to different studies like Miller et al. (2015).

**3  Minor comments**

- **P2, 22:** "... for example present in **the** Antarctic"
- **P2, 37:** Could you give an actual optical thickness range for *high*?
- **P3, 22:** "... using **a** co-located ..."
- **P5, Eq 7:** Your volume depolarization is also a function of $\theta_i$, so $d(R, \theta_i) = \ldots$
- **P6, 9:** "If randomly orientated ice crystals (ROIC) are observed, diattenuation will be strictly $D = 0$ and the scattering Mueller matrix simplifies to a function of two elements,

depolarization d and the volume backscatter coefficient $\beta$." This is not obvious to a non-familiar reader and would need a citation or a descriptive explanation.

– **P7, 30:** Where is the problem, when the system dynamic range is in the "order of 4 to 5 orders of magnitude" and the two polarization signals can differ between "2 orders of magnitude" or "a factor of 2"?

– **P7, 20:** Incomprehensible sentence: "Depolarization effects related to saturation *couple* polarization measurements with terms in the SVLE like cloud base height, range, and optical thickness through signal intensity measurements." What do you mean with "couple"?

– **P3, 22:** "... or the system is insensitive to orientation (as is the case within a few degrees of zenith or nadir) ..." Why is this the case? Could you explain this statement in a little more detail?

– **P8, 30:** "... were accordingly changed ..." → "... were changed accordingly ..."

– **P9, 32:** "... from vertical." → "... from zenith."

– **P9, 11:** "this allows **8** methods to invert and solve Eq. 4." What do you mean by that? What/Are there consequences of having 8 different methods?

– **P9, 18:** "These scans are parsed by *like* polarizations ..." What are *like* polarizations?

– **P10, 3ff:** Could you introduce an explanation of the "opposite sensitivity" of $D_1$ and $D_2$ to saturation for the readers which are unfamiliar with these kind of measurements?

– **P11, 5:** "As a final check, data that is classified as clear air must have substantial signal ..." This is too vague, please give actual threshold.

– **P18, 14-19** Unfortunately, this paragraph is a little hard to follow but key to understand the differences between CAPABL and MPL. Could you rewrite it using an illustrative example?

- **P19, 14:** "... has been pushed ..." → "... has been interpolated ..." (please change this phrase throughout the text)

- **P19, 14:** "... are filtered conservatively" Please elaborate!

- **P21, 31ff:** Can you refer (Figure, Table) from where you take the given percentages?

- **P21, 18:** "An analysis of ground- and space-based observations of HOIC strongly indicates differences based on viewing orientation." Please give a reference.

---

## Referee Comment (RC3) · Anonymous Referee #3 · 21 Nov 2017

Fiedler and Baumgarten (2012) have shown that the importance of understanding the relationship between lidar sensitivity and findings from a time series, e.g. of cloud occurrence. The paper tries to shed new light on this topic. But to my point of view, the authors focus on the wrong issues. It is common knowledge in the lidar community that statistics from analog and photo counting signals are not comparable due to their different sensitivity and dynamic range. For a comprehensive analysis, both signals should be combined (i.e. glued) and not considered separately. In addition, special care needs to be taken to avoid signal saturation. This is usually done by either decreasing laser power or adding neutral density filters into the receiver setup to decrease the strength of the return signal. Saturated signals should not be used at all for data analysis. Finally, it is known that cloud statistics derived only from lidar measurements are biased as a result of signal attenuation by low clouds.

The authors describe that the lidar system they use has been updated from a purely photo counting system to a combined analog and photo counting system. These changes have an effect on the derived cloud phase classification: different statistics of cloud phase classification are found from measurements before and after the update. It would be good to see if these changes are significant. The authors introduce a new algorithm to combine analog and photo counting but the paper comes short in comparing the new algorithm to common methods (e.g. signal gluing). The paper requires some work and can be published after major revisions in AMT.

Major comment:

The comparison presented in Section 3.4 is rather pointless since it cannot be expected that the use of analog and photon counting signals with their different dynamic range will yield the same result. It would be of stronger scientific interest to show that measurement statistics from before and after the system update are consistent. For this comparison, only profiles with unsaturated signals should be used. Without a presentation of measured profiles, it is not possible to assess the reliability of the photo counting channel close to the ground. For example: in Figure 2 (photo counting) below 500 m the depolarization parameter (d) is larger than 0.4 and the backscatter ration (R) close to 0 while in the analog system d is around 0.1 and R around 5. Due to the colour scale is is really hard to give exact numbers but clearly the results are different. In both cases the observations are classified as clear sky but d suggests that in case of photo counting a small number of ice crystals are present compared to the analog signals. Isn't it more likely that the photo counting signal is saturated at lower altitudes. Further, looking at the analog signal close to the ground ( $\sim$  below 100m) d is always larger than 0.5. Are these height bins trustworthy? What is the overlap of your system? How does the profile look like?

AMTD
Section 4, page 4, line 11 to 16. I disagree with the statement that gluing is impractical. It is widely used in the community and does not need atmospheric calibration. Licel provides software for cluing analog and photo counting signals together with their transient recorders. Why do the authors not use this method? There should be no jumps between the analog and photon counting signals (Figure 4) when the gluing is done properly. Can you please show examples of glued profiles and compare the findings to you method? Further, detailed statistics should be provided between the standard gluing method and your new method. What is the effect of using different methods on the cloud phase classification?

Section 5. Comparison of CAPABL to the MPL. Can you assume that the cloud phase classification scheme can be used for the MPL as well? What is the error of d in the MPL? Since the MPL is photo counting only, what did the comparison between MPL and CAPABL look like before the update? Also, how did the CAPABL classification compare to the MWR and MMCR before the update? Are there significant differences to the findings with the new system?

Section 6.2. The authors need to show that the difference between merged signal and analog/photo counting is significant. Why do you use median values and not mean? Is there a big difference between both values?

**Minor comment**

- Due to the pure technical aspect of the paper I would recommend to omit the first paragraph of the introduction. The introduction should focus on lidar systems (i.e. how many system operate in analog and photo counting mode or only use one of these) and lidar analysis (i.e. what methods are used for phase classification – especially in connection to counting system). Also the statement on page 3, line 5 to 9, needs to be discussed in more detail and citation should be provided. - Page 6, line 16, please include e.g. before the citation - Page 7, line 10, How was the transmitter and receiver polarization purity measured? - Page19, line 5 to 8, since the LWP is a column
integrated value: Do you do the comparison when only one cloud layer is observed? What would you expect when you have e.g. a liquid cloud close to the ground and an ice cloud above? - Page 19 line 10, how do you assign LWP when you have more than one cloud layers present? - Page 21, line 18 to 24, HOIC observation are not shown in the paper and entire section should be omitted

Fiedler, J., Baumgarten, G., 2012. On the relationship between lidar sensitivity and tendencies of geophysical time series. In: Reviewed and Revised Papers at the 26th International Laser Radar Conference, Porto Heli, Greece, pp. 63–66.

---

## Author Comment (AC1) · 22 Dec 2017

**Author's Response to Reviewers Comments (Review 1)**
**Title:** Improved Cloud Phase Determination of Low Level Liquid and Mixed Phase Clouds by Enhanced Polarimetric Lidar
The authors would like to first thank the reviewers for their thoughtful and constructive comments on our manuscript. The comments, taken from the provided reviews, have been copied in bullet format and addressed in the sub-bullets. A draft of the changes is also included where omissions are marked with  and additions are given in blue. The line numbers for comments are referenced to the original draft and for responses to the revised draft.

- Comment (Location in Original Draft)
  - Response (Location in Revised Draft)

The authors have changed several portions of this manuscript in line with major changes requested by the reviewers. Section 2 has been reorganized and greatly reduced by including much of the derivation of volume diattenuation and depolarization to Appendix A. A response pertaining to the selected lidar ratios is presented here referring to multiple comments from reviewers. That response is split and referenced as necessary. General clarifications on scope and wording have also been made throughout.

Reviewer 1
- Major Comment 1: To what extent would the results change with different lidar ratios?  Was there a reason such a low lidar ratio was used? (Section 3.2).
  - The goal of this classification is not to quantitatively measure cloud/aerosol backscatter coefficient but rather to differentiate clear air from aerosol and clouds. The lidar ratio and thresholds used to differentiate clear air, aerosols, and cloud particles are linked; changing one will necessitate the change of the other, i.e. raising/lowing the lidar ratio affects the final backscatter ratio from the measurements. However, running our 6-month case study with a fixed set of thresholds and changing the lidar ratio is possible. The results are shown below in Figure R1. Lidar ratios of 10 (solid lines), 20 (dotted lines), and 30 (dash-dotted lines) are presented.
    There is a rule in the classification scheme that overwrites aerosol as ice with high depolarization ratio. This tends to limit the effect of changing the lidar ratio for ROIC and HOIC. Changing the lidar ratio does however start to change the interpretation of liquid and clear air shown by the dramatic reduction of cases labeled by CAPABL as liquid with nonzero LWP. The authors have found that a ratio of 10 is a reasonable threshold to separate liquid from clear air based on the comparison with LWP and radar. The radar comparisons of reflectivity in particular suggest that a lidar ratio of 10 is reasonable, as clear air should have, by far, the lowest reflectivity. Using a higher value for the lidar ratio decreases the number of cases identified by CAPABL as liquid that have non-zero LWP (CDF is shifted up) and raises the number of cases identified by CAPABL as clear with higher reflectivity (CDF is shifted down). With the data presented in Figure R1, we can tune the inversion parameters used to maximize data consistency with non-lidar instrumentation over long periods of time (months to years) and then verify they are reasonable for shorter periods (minutes to hours). Thus, because we only seek to separate these 4 categories (and only because we don't seek to make quantitatively correct determinations of backscatter coefficient), the low lidar ratio (that is constant in time) is reasonable.

[Figure]

**Figure R1: Multisensor analysis of the presented data (manuscript Fig. 6) with differing lidar ratios. The data are presented as described in the legend with line types: Solid lines = lidar ratio of 10, Dotted lines = lidar ratio of 20, and Dash-dot lines = lidar ratio of 30.**

- Major Comment 1: The lidar ratio citation should be to the original authors and not as compiled by Nott and Duck. (Section 3.2)
    - The authors have cited the Hoffmann et al. 2009 paper identified by Nott and Duck. This change can be found on page 9, line 23.
- Major Comment 2: The region where the sharp shifts in FO of liquid and ice at low LDR are relevant, and I think an analysis of this region, and how much changing the thresholds would affect the results, should be presented here (or in a supplementary material /Appendix). (Section 3.3)
    - The authors have performed the analysis suggested by the reviewer. The authors have re-analyzed the entire data set presented with depolarization thresholds from 0.05 to 0.30 with 0.01 spacing. Plotted below, in Figure R2, is the fractional occurrence of liquid and ice measured for July-October 2015 from the analog detection channel. Above approximately $\delta_O = 0.11$, the fractional occurrence stabilizes until approximately $\delta_O = 0.20$. Beyond that point, ice clouds are being lumped into the water fractional occurrence. Any value $0.11 \leq \delta_O \leq 0.20$ will yield similar conclusions for fractional occurrence change. From this we conclude that $\delta_O = 0.11$ is a reasonable threshold to use for the CAPABL data set and based on available literature.

[Figure]

**Figure R2: Fractional occurrence estimates of the first 4 months of available data with varying depolarization ratio thresholds from 0.05 to 0.3 in step sizes of 0.01.**

- Major Comment 3: This analysis seems to underestimate liquid amount. Is it enough to have a single voxel after filtering for a column to be treated as liquid?
    - For this analysis, the authors have considered it sufficient for a single liquid voxel to describe a whole column. Even so, the reviewer is exactly right about underestimating liquid. This classification does tend to underestimate liquid amount because of a number of factors. First, full column measurements of optically thick clouds are, to the authors' knowledge, not currently demonstrated. Lidar systems do however provide a reasonable understanding of cloud base and bottom-of-cloud phase before signal extinction. Second, the unknown lidar ratio as mentioned is a substantial problem for quantitative studies. Finally, reviewer 2 mentions multiple scattering, which is altering the interpretation of approximately 2% of liquid/ice voxels in this data set (seen in Table 5, cell G and described in more detail in the response to reviewer 2's comment). With all of these effects, observing the whole column of liquid remains an issue that lidar systems alone show little promise to completely solve. However, with polar clouds in particular, the ice phase having lower optical thickness than liquid allows for the potential to identify liquid by a very small number of voxels at cloud base. To what extent this facilitates improvement in observing polar clouds can be seen, for example, in Figure 6. 30% of liquid layers that CAPABL identifies are not definitively identified by MWR. This observational bias would result in a skewing cloud radiative effect analyses using LWP information solely from MWRs towards clouds with high optical depth. For deeper analyses of mixed phase clouds, the authors would be remiss to suggest use of this column classification as the only data point, however.

- Major Comment 4: Is there a blind zone between analog and photon counting near 1.5 km? A missed detection analysis is needed to confirm that hydrometeors are not missed by CAPABL to strengthen the reliability of the studied methodology. For example, provide the percentage of occurrence where MWR data has non-zero LQP but CAPABL does not see liquid. Additionally, provide the percentage of occurrences where the MMCR has data above its detection threshold where CAPABL lacks data.
    - The authors agree that such an analysis is needed and now have included the suggested values in the text. Using liquid water path as the indicator of liquid water, CAPABL observes water columns or obscured data at 83% of the times that the MWR observes LWP values greater than its error limit. The MWR observes LWP values greater than its error limit for 3% of clear air/aerosol columns, 10% of ROIC columns, and 4% of HOIC columns. It is important to note, however, that this might be anomalously high for HOIC and ROIC based on errors in LWP retrievals as described below in response to Reviewer 3's Minor comment 5. CAPABL has valid data (passing all the filtering steps described) 75.3% of the time where the MMCR has data above its error threshold. These values have been added to the text in Section 5.3 on page 18, lines 34-35 and on page 19, lines 1-4.
- Major Comment 5: The MPL does not appear to be a great instrument to validate CAPABL. I suppose that the delicate fixed liquid/ice determination thresholds have a more significant role in the MPL data analysis as well. (Section 5)
    - The authors need to clarify that the analysis we termed multisensory validation is actually a comparison. The title of Section 5 and subsequent references have been changed to reflect this clarification. There are a number of reasons that comparison with the MPL is of concern. This is one reason to first include the analysis of analog, photon counting, and merged data in Figures 3 and 5 from CAPABL data alone. We consider the analysis an internal comparison that indicates that low-level Arctic clouds are extremely difficult to observe fully with purely photon counting detection. This calibrates expectation on the later comparison between CAPABL and the MPL. This internal comparison is less sensitive to the actual threshold values described in Table 2 than the comparison between CAPABL and the MPL, indicating the issues related to inability to fully measure the dynamic range of interest. We too are surprised by the overall performance of the MPL but after completing the comparison of CAPABL's photon counting signals with our fully merged data product, we believe the results to be aligned with the overall capability.
- Minor Comment 1: Add "volume pixel" in parentheses after "voxel". (Page 9, Line 25)
    - The suggested change has been made. This can be found now on page 8, line 33.
- Minor Comment 2: Change "is" to "are". (Page 14, Line 9)
    - The suggested change has been made. This can be found now on page 12, line 29.
- Minor Comment 3: Please be consistent and use either "Fig." or "Figure". (Page 15, Line 2 and throughout)
    - The guidelines followed were to use "Fig." in running text and "Figure" at the start of sentences. Consistency changes have been made in the following places but not in the line identified by the reviewer:
        - Page 13, line 19
        - Page 18, line 13
        - Page 18, line 18
        - Page 18, line 24
        - Page 19, line 8
        - Page 20, line 33
- Minor Comment 4: Please provide a citation for this low LWP uncertainty. (Page 16, Line 27-28)
    - The Cadeddu et al. 2013 reference is appropriate here and has been moved from the

previous line (Page 16, Line 29) to the end of the cited information. The new citation can be found on page 15, lines 21-22.

- Minor Comment 5: Change "has" to "have". (Page 19, Line 17)
  o The suggested change has been made. This change can be found on page 18, line 13.
- Minor Comment 6: Please provide a citation for this argument regarding the cirrus mode artifacts. (Page 20, Line 2-3)
  o The Clothiaux et al. 1999 reference is added to page 19, line 9.
- Minor Comment 7: Could there be a height effect of CAPABL measurements as well affecting the integrated column measurements?
  o The authors believe this question refers to our analysis of lower radar reflectivity values for HOIC vs. ROIC. If so, there is almost certainly a geophysical height effect to the observations of HOIC related to the ambient temperature. Much work related to the CALIOP instrument aboard the CALIPSO satellite (e.g. Noel and Chepfer (2010)) indicates strong temperature dependence based on ice crystal habit. That said, having the ability to better sample a region of the atmosphere, in this case low altitudes from ground based measurements, almost certainly causes non-ideal sampling (though to the authors' knowledge not currently provable). It is entirely possible that HOIC are much larger on average than ROIC (indicating they should have higher radar reflectivity) and that the population observed to create our presented results have a bias. The authors simply intend to call the readers attention to this suspected bias as they related to satellite based measurements that can be similarly biased but to observe high, rather than low, clouds.
- Minor Comment 8: Please consider changing the colorbar around 0 to grey, as it is impossible to distinguish between missing or "bad" lidar returns and values near zero. (Figure 1)
  o The suggested change has been made. The colorbars on Figure 1 and 2, subplots Diattenuation and Backscatter Ratio, have had white removed from them.
- Minor Comment 8-9: Consider extending the scale of the relative backscatter panel, as it is impossible to separate the intense lower liquid layers' returns from the noisy background. (Figure 1 and Figure 4).
  o The colorbars on Figures 1, 2, and 4 have all been extended as suggested.
- Minor Comment 8: I suggest adding daily sounding temperature profile to the plot to enable the examination of the reliability of the HOIC classification given the temperature range. (Figure 1)
  o The authors believe that a complete analysis of the temperature dependence of our HOIC measurements would be extremely interesting and certainly enlightening. The author have elected however to leave the temperature dependence off of our Figures 1 and 2 in favor of the measurements used in our classification scheme. The reasons for this are as follows:
    1. The authors make no requirement of the HOIC flag based on temperatures. Though findings from CALIOP aboard the CALIPSO satellite (e.g. Noel and Chepfer (2010)) indicate temperature dependence, we do not leverage it for our classification scheme. We thus find that adding temperature to Figures 1 and 2 would distract from the flow intended with these figures and with Table 2.
    2. The authors are currently performing an analysis of the HOIC flag relative to all ancillary measurements at Summit. The temperature of instances of HOIC measured from radiosondes are given below in Figure R3. This analysis is found to be by no means simple and is well beyond the scope of this paper, in which the authors really want to focus on non-orthogonal polarization measurements and classification.

[Figure]

**Figure R3: Temperatures of ROIC and HOIC voxels interpolated from radiosonde measurements for the first year of available CAPABL data.**

- Minor Comment 10: Please have a citation for the approximation given in the caption. (Figure 6 caption)
  - A reference to Bendix (2002) has been added as suggested.

Reviewer 2
- Major Comment 1: Several passages should be moved throughout the text to better organize your work and motivate your lengthy derivations in Section 2.1.
  - The authors agree with the reviewer that motivation is required for the more technical derivation portion of this work. We have taken the suggested paragraph and moved it to the front of section 2.1. This can now be found on pages 3, lines 28-33 and page 4, lines 1-7. The suggested summary sentence about the enhancements due to non-orthogonal polarization measurements and analog/photon counting detection has been added to the abstract. This change can now be found on page 1, lines 14-15.
- Major Comment 1: Drastically reduce the equations in part 2.1.
  - The suggested change has been made with derivation equations now included in Appendix A for the interested reader. The remaining equations are the Stokes vector lidar equation from which all variables are defined, the definitions of volume depolarization and volume diattenation used for this classification scheme, and the criteria required allowing valid inversions when using a 3-polarization inversion method. The authors have thus moved all the detailed definitions and inversions for the curious reader while maintaining the relevant equations.
- Major Comment 2: Can you elaborate on your decision to use a lidar ratio of 10 and/or check if your results change for other lidar ratios values?
  - The analysis requested has been provided above in Figure R1 in response to a similar comment by Reviewer 1 (major comment 1). The value of the lidar ratio does not change the analysis provided of ROIC/HOIC because of the classification rule that overwrites aerosol layers with high depolarization ratio values as ice. This additional rule softens the hard limit indicated by the classification thresholds. The fixed lidar ratio value does affect the liquid/clear air interpretation, however, and is set to 10 for this analysis. This value is a direct result of the desire not to quantitatively assess cloud backscatter coefficients but rather to separate clouds from clear air and aerosol layers. This limitation of not knowing the lidar ratio in the observations made by elastic scattering lidar systems necessitates the restriction in scope of scientific inquiry that is possible to unambiguously address. The authors believe that the goal of classification is well within this scope and demonstrate in our manuscript logical consistency with instrumentation that does not require such assumptions.
- Major Comment 3: Can you quantify the difference that multiple scattering causes in your data set using the two fields of view you describe?
  - The authors agree that multiple scattering should be addressed. We have added the requested comparison, which comes directly from Table 5. In the summer months, up to 5% of liquid voxels identified by the MPL are mischaracterized by CAPABL; this value falls to 2% over the 6-month period. These values have been added to the text in Section 5.2 on page 17, lines 18-24.
- Major Comment 4: Suggest removing the subplots for clear air and ice voxels and focus primarily on the liquid panel in Figure 3.
  - The authors have followed the reviewer's suggestion and removed the ice and clear air portion of Figure 3. The authors have also modified the text on page 11, lines 22-26 accordingly to focus on liquid voxels. The ice and clear air portions have been relocated to Appendix C. The authors feel that this relocation is reasonable because the main point of voxel misidentification is still made and the interested reader can still find a complete analysis if curious.
- Major Comment 4: Please provide PDF versions of Figures 6 and 7 to decide if cumulative distribution functions are the best visualization of the data.

o The requested figures are included below (Figures R4 and R5 correspond to manuscript Figure 6 and Figures R6 and R7 correspond to manuscript Figure 7). The attached probability density functions are normalized by their area to 1 (Figures R4 and R6) and to their maximum value to bring all functions into the same scale (Figures R5 and R7). Note that with the normalization to the maximum value, the PDFs do not have an integral of 1 but the scales are the same for comparison purposes. The authors prefer CDFs for a number of reasons:

1. PDFs represent probability with areas while CDFs represent probability with vertical distances. Because the authors are trying to represent a fundamentally continuous distribution with sampled data that is discrete, the data is not smooth. It is clearer to see the changes in vertical distance on the CDFs than to estimate the area of a jagged object.

2. CDFs are independent of the number of histogram bins used to create them where PDFs can change their behavior (how jagged they are) based on the width of bins chosen again due to the jagged nature of sampled data.

3. End cases appear cleaner in CDFs. Two possibilities exist for end cases with PDFs: either all end cases are lumped in the first or final bin resulting in erroneous spikes or end cases are not observed giving the false impression that the integral areas represented are necessarily equal. For example, in Doppler Velocity or radar SNR (Figure R4 and R5 below), misleading spikes appear because the authors have chosen to include all data. These spikes can be wrongly interpreted as important when they are simply data collected from all end cases and do not have the same bin width as other data shown.

4. The CDF will by definition always have the same bounds (0-100%) but a PDF that requires a strict integral of 1 has the issue that scales of broad and narrow distributions can be sharply different (such as the LWP plot below).

[Figure]

**Figure R4: Probability density functions (area normalized to 1) of the multisensor comparison presented in manuscript Figure 6.**

[Figure]

**Figure R5: Normalized probability density function (normalized to the maximum value) of the multisensor comparison presented in manuscript Figure 6.**

[Figure]

**Figure R6: Probability density functions (area normalized to 1) of the radiation comparison presented in manuscript Figure 7.**

[Figure]

**Figure R7: Normalized probability density function (normalized to the maximum value) of the radiation comparison presented in manuscript Figure 7.**

- Major Comment 5: A comparison of absolute values from cited literature of downwelling radiative effects would be helpful.
  - The authors feel that this is beyond the scope of this work. The radiation data presented is not directly comparable to Miller et al. (2015), which conducted an analysis of cloud radiative forcing/effect at Summit from 2011 to 2013. Because Miller et al. (2015) calculated cloud radiative effect instead of using raw radiation measurements the results are not directly comparable. The difference in these analyses includes the difficult removal of clear air radiative effect that is, in the authors' opinion, well beyond the scope of this paper, which aims to demonstrate a lidar retrieval and classification method rather than diagnose the impact of clouds on the surface. This is planned for future analysis, however.
- Minor Comment 1: Insert "the" in the sentence: "…for example present in Antarctic". (Page 2, Line 22)
  - The suggested change has been made. This change can now be found on page 1, line 22.
- Minor Comment 2: Could you give the actual optical thickness range for "high"? (Page 2, Line 37)
  - The authors have clarified in their footnote number 2 on page 2 that OD is considered high around OD 5.
- Minor Comment 3: Insert "a" in the sentence: "…is presented in Sect. 5 using co-located micropulse lidar…". (Page 3, Line 22)
  - The suggested change has been made. It can now be found on page 3, line 23.
- Minor Comment 4: Volume depolarization is a function of observation angle. (Equation 7)
  - The suggested change has been made in Equation 7 (now 2) and Equation 8 (now 3) on page 5, lines 2 and 4.

- Minor Comment 5: The sentence "If randomly orientated ice crystals (ROIC) are observed, diattenuation will be strictly D = 0 and the scattering Mueller matrix simplifies to a function of two elements, depolarization d and the volume backscatter coefficient β" requires a citation as a non-obvious fact. (Page 6, Line 9)
    - A reference to Hayman and Thayer 2012 has been added on page 5, line 24.
- Minor Comment 6: Where is the problem, when the system dynamic range is in the "order of 4 to 5 orders of magnitude" and the two polarization signals can differ between "2 orders of magnitude" or "a factor of 2"? (Page 7, Line 30)
    - The issue here is detailed in Section 6.1. Large differences in dynamic range caused by polarization measurements fundamentally limit the altitude range of measurements (especially the perpendicular signal), i.e. there are too few photons in the perpendicular channel. In doing so, limiting the altitude range of measurements limits the validity of measurements when trying to use them to attribute other effects like cloud radiative effect. In the case of CAPABL's photon counting channel, this limit causes a misrepresentation of approximately 1/3 of liquid clouds. In the case of analog detection, somewhere between 4 and 22% of high ice clouds are missed.
- Minor Comment 7: Incomprehensible sentence: "Depolarization effects related to saturation *couple* polarization measurements with terms in the SVLE like cloud base height, range, and optical thickness through signal intensity measurements." What do you mean with "couple"? (Page 7, Line 30)
    - The authors have modified the original sentence to clarify the "coupling" is really a link between macro and microphysical properties. The new sentence reads: "Depolarization effects related to saturation link polarization measurements (microphysical properties) with properties like cloud base height, range, and optical thickness (macrophysical properties) that have a strong influence on the signal intensity of the measurements." This change can be found on page 4, lines 2-4.
- Minor Comment 8: Why is the following sentence true? "…or the system is insensitive to orientation (as is the case within a few degrees of zenith or nadir) ..." (Page 3, Line 22)
    - Orientation is identified by the diattenuation flag. However, the measured diattenuation of the volume is the weighted average of the diattenuation of ROIC and HOIC, weighted by the occurrence frequency and scattering efficiency. ROIC should show zero diattenuation while HOIC show diattenuation that is a strict function of observation angle. The sensitivity of CAPABL to diattenuation peaks near 32 degrees (its current tilt angle) based on the strong increase of backscattering efficiency from oriented plates due to the corner reflection in ice. Given that HOIC are expected to be a small overall fraction of ice crystals, strong diattenuation (which can not necessarily be expected) or strong backscattering from HOIC are required to make the oriented fraction of the voxel dominate the randomly oriented fraction. Within a few degrees of zenith/nadiar (for example the current tilt angle of CALIOP is 3 degrees) there is no strong diattenuation nor is there strong enhancement in backscattering efficiency from HOIC. However, identifications by instruments like CALIOP operating at an angle close to 0.3 degrees see enhanced scattering efficiency from HOIC as well, making the HOIC identification less about the observed diattenuation and more about signal strength. The authors have included this point in footnote 3 on page 7 to clarify the point while maintaining the focus of the sentence on the desired effect of saturation.
- Minor Comment 9: Change "…were accordingly changed…" to "...were changed accordingly". (Page 8, Line 30)
    - The suggested change has been made and can now be found on page 8, line 4.
- Minor Comment 10: Change "…from vertical" to "from zenith". (Page 9, Line 2)
    - The suggested change has been made and can be found on page 8, line 6.

- Minor Comment 11: What do you mean in the following sentence: "this allows 8 methods to invert and solve Eq. 4"? What are the consequences of having 8 different methods? (Page 9, Line 11)
  - o The section has been modified to clarify that the 8 inversion methods facilitate flexibility when trying to retrieve cloud properties. If for example the parallel channel is subject to saturation, it need not necessarily be used to retrieve cloud properties. Likewise, for high thin clouds the perpendicular channel, which is typically too weak to make reliable measurements, need not be used. This can be found on page 8, line 18-19.
- Minor Comment 12: What are "like" polarizations? (Page 9, Line 18)
  - o Data is taken by scanning polarizations as $\theta_1, \theta_2, \theta_3, \theta_4, \theta_1, \ldots$ and saved sequentially. This step simply unpacks raw data and splits the data stream. This sentence has been removed, as it doesn't add clarity to the description.
- Minor Comment 13: Could you introduce an explanation of the "opposite sensitivity" of $D_1$ and $D_2$ to saturation for the readers that are unfamiliar with these kind of measurements? (Page 10, Line 3ff)
  - o The sentence described has been changed to describe the opposite sensitivity as opposite sign value in bias. This can be found on page 6, line 29-30.
- Minor Comment 14: The following sentence is too vague: "As a final check, data that is classified as clear air must have substantial signal…". Give the actual threshold. (Page 11, Line 5)
  - o An additional sentence has been added clarifying that this classification scheme requires greater than 66% data availability from 1-2 km to be considered clear air. This can be found on page 10, lines 20-21.
- Minor Comment 15: Can you rewrite the paragraph beginning on Page 18, Line 14 to be clearer? Suggest using an illustrative example. (Page 18, Lines 14-19)
  - o The authors agree with Reviewer 2 and have changed the paragraph to the following: "The data presented in Table 5 for December observations shows a large disagreement between CAPABL and the MPL (Table 5 cell D). Here CAPABL data fails QC filtering but MPL data is classified as clear air. The majority of the CAPABL observations filtered from the analysis are excluded because they do not meet the requirements of being a valid diattenuation observations. Either, the measurements do not pass the consistency test or have an unacceptably large error. Because the diattenuation filtering is unique to CAPABL, applying this exact filtering scheme to the MPL is impossible and CAPABL data is filtered more conservatively than the MPL given the same bounds for filters common to both instruments." This text may now be found on page 17, lines 7-12.
- Minor Comment 16: Change "…has been pushed…" to "…has been interpolated…". (Page 19, Line 14 and throughout the text)
  - o The suggested change has been made of the following lines:
    - ▪ Page 15, line 12
    - ▪ Page 15, line 22
    - ▪ Page 18, line 14
- Minor Comment 17: Please clarify the following sentence: "The values given in Fig. 6 are filtered conservatively". (Page 19, Line 33)
  - o This sentence has been changed to indicate that conservative filtering is in the column data mask described in Section 3.3 and that this filtering results in approximately 5% of data that should be labeled "Obscured" that is allowed to be labeled "Clear Air". This change can be found on page 18, line 30-32.
- Minor Comment 18: Can you refer to the Figure or Table for where you take the given percentages? (Page 21, Line 31ff)
  - o The results in the section described now include a direct statement describing the origin of the results. The changes can be found on page 21, lines 15 and 19-20.

- Minor Comment 19: Provide a reference for the following sentence: "An analysis of ground and space-based observations of HOIC strongly indicates differences based on viewing orientation". (Page 21, Line 18)
    - o The sentenced referenced has been removed in response to a comment by reviewer 3. As such, no citation has been included.

Reviewer 3

- Major Comment 1: The comparison in Section 3.4 is rather pointless. It would be of greater scientific value to demonstrate the profiles from before and after the hardware update are consistent.
    - o The comparison in Section 3.4 is intended as a direct example of the mechanism of saturation causing changes to the geophysical interpretation of our classification strategy. The primary goal of this comparison is not to describe the instrument upgrade and continued development work. Rather, it is to demonstrate the capability of the classification scheme of our current system setup. Furthermore, without the internal comparison between analog and photon counting, the comparisons of CAPABL's merged data product and the MPL data lacks critical context. Specifically how well CAPABL's own photon counting data matches the overall merged data product informs the expectation of how well a different system's photon counting data should match the overall data product.
    The reviewer is correct that a full presentation of the entire data set for CAPABL would be of great scientific interest. Unfortunately, before July 2015, major research and development work prevented continuous measurements. In addition to the highlighted hardware changes, major changes were made to CAPABL's operational software and post processing methods to increase reliability and stability. This combined with major hardware failures in the winter, where the authors were unable to perform repairs, precluded high quality continuous measurements. Before July 2015, the operational state of CAPABL is not really comparable to its current operational state. The authors therefore concluded that such a comparison, while extremely interesting, is not possible. For these reasons, the authors feel that the comparison presented in Section 3.4 should remain as presented in the current version of the manuscript.
- Major Comment 2: Low altitude data looks untrustworthy? What is the system overlap and how do your data look?
    - o Indeed, the low altitude regions are extremely difficult to measure accurately. This is part of the motivation of this study to push measurement reliability lower. Without the addition of a dedicated low altitude channel, which is impractical at Summit given the severe bandwidth limitations of data transfer from the site, the lowest altitudes are the hardest to measure well. The system overlap is shown below in Figure R8. This is calculated using the system parameters using standard ABCD matrices.

[Figure]

**Figure R8: CAPABL's overlap function as a function of range.**

In clear air, data above approximately 100 meters is quite reliable but not within clouds. Low fog or clouds almost always result in data clipping, which is flagged and removed by the Licel counting system and operational software. This is seen, for example, near noon in Figures 1 and 2 and from noon to 17 UTC in Figure 4. The data is flagged and removed that looks suspiciously like a low cloud but given this clipping, quantitative assessment is impossible. The Klett inversion used can be unreliable below approximately 100-150 meters as the system overlap correction becomes quiet large. As a practical matter of the comparison with other sensors, the MPL has the same problem and is in fact worse because the lack of an analog detection channel and additional noise caused by signal induced noise resulting from the use of a transceiver design. The radar also has a blind zone due to its transceiver and pulse length resulting in data that is unreported. Finally, the obscuration flag in the column data product is a direct result of comparisons with MWR and trying to remove data that is clearly unreliable.

- Major Comment 3: I disagree with the statement that gluing is impractical. Why do the authors not use the method? Can you please show examples of glued profiles and compare findings to your method?
    - The authors' had intended to say that the use of gluing is impractical for our particular application of making observations at Summit and not as a general statement. The sentence has been revised to: "it is impractical to calculate gluing coefficients for CAPABL by atmospheric calibration as access to the CAPABL system is limited to once or twice a year". Of particular concern to the authors is the temporal variability of the gluing coefficients on the scale of weeks to months. The analysis of Newsom et al. (2009) indicates diurnal variability in particular. We are granted limited access to the system in the summer at the same time of year each year. As such, the authors have no ability to test any possible variability due to seasonality or background conditions (the sun is always above the horizon during our access period). This comment has been clarified within the text to specify that it is not a general statement but rather a limit of the

access of a remote site like Summit. With this inability to test temporal variability of the gluing coefficients sufficiently, we did not provide gluing products of any type out of an abundance of caution and resist the request to do so, due to our incomplete understanding of the temporal variation specific to CAPABL.

An additional concern for gluing at a site like Summit is the physical nature of the clouds. It is the authors' experience based on nearly 7 years of data from ICECAPS that clouds create a bimodal distribution in peak signal strength caused largely by optical thickness and base height. Low optically-thick liquid clouds almost exclusively cause extreme saturation in photon counting data and higher ice clouds are well represented in photon counting data. This behavior in addition to the issues highlighted in Section 4 force us to combine data at the product level instead of the raw signals. This is not burdensome in our case, as we already must combine data at the product level to perform the data merging for different polarization retrieval angles.

- Major Comment 4: Can you assume that the cloud phase classification scheme can be used for the MPL as well?
  - o The authors based the development on the presented scheme for CAPABL on the literature for polarization lidars. This includes papers focused on analysis of data from MPLs such as those of Campbell et al. (2002) and Flynn et al. (2007). Additionally, the multisensor cloud phase classification paper by Shupe (2007) is also a major basis for this work. The applicable rules and processing steps described in Table 2 are thus expected to easily apply to the MPL. The corrections described by Campbell, including afterpulse calibration, overlap calibration, and saturation corrections, are also required for the MPL. Beyond the specifics of those calibration procedures, polarization analyses based on diattenuation measurements are not possible with the MPL. Subject to those differences, the assumption that this scheme would be valid was made given the literature. The results in Table 5 show encouraging signs that this scheme works well for the MPL (specifically high values in cells A and P) with the noted exception in the text that the MPL seems to miss cloud cases based on noisy data causing errors in the Klett inversion procedure.
- Major Comment 5: What is the error of d in the MPL?
  - o The authors follow the derivation of depolarization given by Flynn et al. (2007) as the derivation we present assumes multiple strictly linear polarizations. Flynn et al. (2007) present a derivation for a standard 2-channel measurement beginning with a relevant version of the Stokes vector lidar equation. A rearrangement of their Eq. 1.5 yields:

$$d = \frac{2N_\perp}{2N_\perp + N_{Circ}}$$

The propagation of error of this expression for the error estimate is:

$$\sigma_d = 2\sqrt{\frac{N_{circ}^2 N_{\perp_T} - N_\perp^2 N_{Circ_T}}{(2N_\perp + N_{Circ})^4}}$$

We use the above expression to calculate error for the MPL where $N_\perp$ and $N_{Circ}$ are the number of background subtracted photons in the perpendicular and circular channels, respectively, and $N_{\perp_T}$ and $N_{Circ_T}$ and the total number of counts, including background, in the perpendicular and circular channels, respectively. From these expressions, the typical linear depolarization ratio is given by Flynn et al. (2007) in their Eq. 1.5 as:

$$\delta = \frac{d}{2-d} = \frac{\frac{2N_\perp}{2N_\perp + N_{Circ}}}{2 - \frac{2N_\perp}{2N_\perp + N_{Circ}}} = \frac{N_\perp}{N_\perp + N_{Circ}}$$

The propagation of error of this expression for the error estimate is:

$$\sigma_d = \sqrt{\frac{N_{circ}^2 N_{\perp_T} - N_\perp^2 N_{Circ_T}}{(N_\perp + N_{Circ})^4}}$$

We calculate this error for every point for use in the classification described in Table 2.

- Major Comment 6: What did a comparison of the MPL and CAPABL look like before the update?
    - The authors have not compared the data from our developmental phase of CAPABL to MPL data in a complete fashion. The purpose of this paper is to demonstrate and test our current enhanced ability to measure Arctic clouds. Of particular interest here is the ability to test the classification scheme and compare it to other operational measurement systems on an operational basis. This ability has been developed and made more robust over several years of testing and development. The dates of comparison were chosen based largely on instrument uptime (CAPABL and other ICECAPS instruments shown in Figure R9 below), which has been dramatically improved for CAPABL from its original installation to present. As such, we consider the first date for comparisons to be July 2015. Before this, we do not posses CAPABL data that is adequately continuous to demonstrate our method.

[Figure]

**Figure R9: ICECAPS sensor uptime for the period of comparison for manuscript Figure 6.**

- Major Comment 7: Demonstrate that the difference between the merged signals and the analog/photon counting is significant.
    - The authors agree with the reviewer that this is an important oversight in our manuscript. Section 6.1 has been modified to include the coverage from just orthogonal measurements (what would be available for just analog and photon counting data without the non-orthogonal components) compared to our full data retrieval. The text now reads: "At Summit, CAPABL provides a fully merged data product that covered 34% of the

column from 0 km to 8 km for July to December 2016. Using only orthogonal components from analog and photon counting results in only 25% coverage. In comparison to CAPABL, the MPL observed 19% of the column above Summit in summer (CAPABL observes 25% for the fully merged mask and only 18% for the orthogonal components) and 44% in winter (CAPABL observes 45% for the fully merged mask and only 31% for the orthogonal components)". This change may be found on page 20, lines 4-9.

- Major Comment 8: Why do you use median values and not mean? Is there a large difference?
  - o Median values of the radiation data are presented, as they are consistent with the presentation of the data as a cumulative distribution function. The qualitative results are unchanged using the mean values.
- Minor Comment 1: Recommend omitting the first introduction paragraph. Recommend focusing more on lidar system.
  - o The authors agree with the reviewer that the focus of the paper should be on lidar measurements and demonstrating the described non-orthogonal retrievals. The authors do, however, have a strong preference to leave the first paragraph of the introduction. We intend to use it as scientific motivation for making measurements of low-level clouds in the Arctic. Without such motivation, the authors fear this work lacks specific impetus to fill observational gaps in the Arctic.
- Minor Comment 2: The statement on Page 3, lines 5-9 needs to be discussed in more detail or citation provided.
  - o References to Biele et al. 2000, Alvarez et al. 2006, and Hayman and Thayer 2009 have been added here. This change can be found on page 3, line 6-7.
- Minor Comment 3: Please include "e.g." before the citation (Page 6, Line 16)
  - o The suggested change has been made and can be found on page 6, line 2.
- Minor Comment 4: How were the transmitter and receiver polarization purity measured? (Page 7, Line 10)
  - o The measurements specified have been performed as follows. The transmitter is measured using the following optical setup: Laser source → polarizer → transmitter optics → analyzer → detector. The polarizer and analyzer are both Glan-Taylor polarizers as specified in Table 1 caption. The receiver is done similarly: Laser source → polarizer → receiver optics → analyzer → detector. The full Stokes vector of the laser source after the polarizer and after the transmitter/receiver optics was measured. The overall degree of polarization was measured before and after to determine rejection/purity. After this initial step, this is verified with the full operational system using clear air atmospheric returns. CAPABL's analyzer uses a liquid crystal variable retarder combined with a quarter wave plate to make a variable rotator whose voltage can be scanned in clear air periods to map voltage values to polarization orientations. These scans are used to verify polarization purity at several altitudes both in the boundary layer and above in the lower stratosphere. The given values are tested each time the system is visited by the authors and verified to be consistent from initial installation in 2015 to current. An example is given in Figure R10 using both of CAPABL's lasers, which are cross-polarized. The voltage of the max/min of each profile is determined in this way as well as the 45 degree points where the profiles overlap. Note that the below profiles are taken during a period of mostly clear air for the gray lines and optically thin blowing snow for the black lines. The overall minimum measurable depolarization is approximately 1% in clear air. One final note to make is that this scan is run at relatively coarse resolution in voltage to speed it for initial polarization determination. Higher resolution scans are run to set actual values once the broad structure is confirmed. This higher resolution is needed near the minima, which are very sharp features.

[Figure]

**Figure R10: Liquid crystal variable rotator (LCVR) scans performed by CAPABL. The voltage applied is not linearly related to the analyzer polarization angle. The full system minimum observable depolarization is determined in this fashion after testing the receiver and transmitter individually.**

- Do you do the comparison of LWP when more than 1 cloud layer is observed? What would you expect with a mixed phase layer? (Minor 5 on Page 19, Lines 5-8 and Line 10)
    - o Yes, the comparison of LWP is done on all data. The process used to retrieve LWP requires 2 radiometers to measure emission from the atmosphere in 3 different microwave bands. Multi-layer and single layer cloud emission is not intrinsically a problem to measure as long as the emission is not scattered by small water droplets. Given the low optical depth in the selected microwave bands, this is a negligible quantity for almost all cases. However, Pettersen et al. (2016) and (2017), for example, show the higher order effects of scattered surface emission caused by larger ice particles within deep ice layers causing non-zero liquid water path observations that actually lack liquid water. The authors hypothesize that this is one reason that ROIC/HOIC layers can have non-zero LWP as shown in Figure 6. Analysis of this impact is planned for future work.

Pettersen, et al., "Microwave signatures of ice hydrometeors from ground-based observations above Summit, Greenland", ACP, 2016, doi: 10.5194/acp-16-4743-2016
Pettersen, et al., "Precipitation regimes over central Greenland inferred from 5 years of ICECAPS observations", ACPD, https://doi.org/10.5194/acp-2017-857, in review, 2017.

- Minor Comment 6: HOIC observations are not shown in the paper and the entire section should be omitted (Minor 6 on Page 21, Lines 18-24)
    - o The suggested change has been made.

[revised manuscript text omitted]

$$\bar{\bar{M}}_{T_x}(\bar{k}_i)\bar{S}_{T_x} = \begin{bmatrix} 1 & \cos(2\phi) & \sin(2\phi) & 0 \end{bmatrix}^T,$$

2) only measures linear polarized signal at angle $\theta$ from the reference transmit polarization, (Eq. 15 with $A(\Gamma_{wp}) = \bar{\bar{M}}_{R_x}(2\theta)$)
20 yielding the simplification

$$\bar{\bar{M}}_{R_x}(\bar{k}_s) = \frac{1}{2}\begin{bmatrix} 1 & \cos(2\theta) & \sin(2\theta) & 0 \\ 1 & \cos(2\theta) & \sin(2\theta) & 0 \\ 0 & 0 & 0 & 0 \\ 0 & 0 & 0 & 0 \end{bmatrix},$$

and 3) using the definition of the backscattering phase matrix

$$\bar{\bar{F}}\left(\bar{k}_i, -\bar{k}_i, R\right) = \begin{bmatrix} F_{11}(R) & F_{12}(R) & 0 & 0 \\ F_{12}(R) & F_{22}(R) & 0 & 0 \\ 0 & 0 & F_{33}(R) & F_{34}(R) \\ 0 & 0 & F_{34}(R) & F_{44}(R) \end{bmatrix} \tag{5}$$

the number of photons to be observed in any arbitrary linear polarization channel is given in Eq. 6 as

$$N_M(R) = \xi(R)\left[F_{11}(R) + \cos(2\theta)F_{12}(R) + \cos(2\phi)\left(F_{12}(R) + \cos(2\theta)F_{22}(R)\right) + \sin(2\theta)\sin(2\phi)F_{33}(R)\right]. \tag{6}$$

5    Here, all constant terms of Eq. 1, which will cancel when taking signal ratios, are lumped into the term $\xi(R)$ such as the measurement solid angle, geometric overlap, range resolution, and atmospheric transmission.

The number of measured photons incident upon the photodetector, $N_M(R)$, is a function of transmitted and received polarization angle $\phi$ and $\theta$, respectively, and is related to the scattering phase matrix terms, $F_{11}(R)$, $F_{12}(R)$, $F_{22}(R)$, and $F_{33}(R)$, which are all functions of range. For CAPABL, $\phi = 45^o$; applying this constraint to Eq. A4 cancels the functional

10    dependency on $F_{22}(R)$ by design. Thus, using three distinct receiver polarization channels: $\theta_1$, $\theta_2$, and $\theta_3$, one can create a set of three simultaneous equations which can be inverted to calculate the Mueller matrix terms of interest that describe backscattering coefficeint ($F_{11}$), volume depolarization ($F_{33}/F_{11}$), and volume diattenuation ($F_{12}/F_{11}$). This set of equations is given in Eq. A5

$$\begin{bmatrix} N_1(R) \\ N_2(R) \\ N_3(R) \end{bmatrix} = \xi(R) \begin{bmatrix} 1 & \cos(2\theta_1) & \sin(2\theta_1) \\ 1 & \cos(2\theta_2) & \sin(2\theta_2) \\ 1 & \cos(2\theta_3) & \sin(2\theta_3) \end{bmatrix} \begin{bmatrix} F_{11}(R) \\ F_{12}(R) \\ F_{33}(R) \end{bmatrix} \rightarrow \bar{N} = \bar{\bar{A}}\bar{F}. \tag{7}$$

15    The general matrix inverse of $\bar{\bar{A}}$ is given in Eq. 8 as

$$\bar{\bar{A}}^{-1} = \frac{1}{\zeta} \begin{bmatrix} \sin(2\theta_2 - 2\theta_3) & \sin(2\theta_3 - 2\theta_1) & \sin(2\theta_1 - 2\theta_2) \\ \sin(2\theta_3) - \sin(2\theta_2) & \sin(2\theta_1) - \sin(2\theta_3) & \sin(2\theta_2) - \sin(2\theta_1) \\ \cos(2\theta_2) - \cos(2\theta_3) & \cos(2\theta_3) - \cos(2\theta_1) & \cos(2\theta_1) - \cos(2\theta_2) \end{bmatrix}. \tag{8}$$

Note that the matrix $\bar{\bar{A}}$ and the matrix inverse $\bar{\bar{A}}^{-1}$ are not functions of range but only of the selected receiver polarizations. The term

$$\zeta = \cos(2\theta_3)\left(\sin(2\theta_2) - \sin(2\theta_1)\right) + \cos(2\theta_1)\left(\sin(2\theta_3) - \sin(2\theta_2)\right) + \cos(2\theta_2)\left(\sin(2\theta_1) - \sin(2\theta_3)\
[revised manuscript text omitted]
\left(R, \theta_i\right) - 1 = \frac{F_{33}\left(R, \theta_i\right)}{F_{11}\left(R, \theta_i\right)} = \frac{\left(\cos\left(2\theta_3\right) - \cos\left(2\theta_2\right)\right) N_1\left(R\right) + \left(\cos\left(2\theta_1\right) - \cos\left(2\theta_3\right)\right) N_2\left(R\right) + \left(\cos\left(2\theta_2\right) - \cos\left(2\theta_1\right)\right) N_3\left(R\right)}{\sin\left(2\theta_2 - 2\theta_3\right) N_1\left(R\right) + \sin\left(2\theta_3 - 2\theta_1\right) N_2\left(R\right) + \sin\left(2\theta_1 - 2\theta_2\right) N_3\left(R\right)} \tag{A8}$$

10 and volume diattenuation,

$$D\left(R, \theta_i\right) = \frac{F_{12}\left(R, \theta_i\right)}{F_{11}\left(R, \theta_i\right)} = \frac{\
[revised manuscript text omitted]